# Pyramidal cell subtype-dependent cortical oscillatory activity regulates motor learning

Takeshi Otsuka [1,2,4] ✉ & Yasuo Kawaguchi [1,2,3]

The cortex processes information through intricate circuitry and outputs to multiple brain areas by different sets of pyramidal cells (PCs). PCs form intra- and inter-laminar subnetworks, depending on PC projection subtypes. However, it remains unknown how individual PC subtypes are involved in cortical network activity and, thereby, in distinct brain functions. Here, we examined the effects of optogenetic manipulations of specific PC subtypes on network activity in the motor cortex. In layer V, the beta/gamma frequency band of oscillation was evoked by photostimulation, depending on PC subtypes. Our experimental and simulation results suggest that oscillatory activity is generated in reciprocal connections between pyramidal tract (PT) and fast-spiking cells. A similar frequency band was also observed in local field potentials during a pattern learning task. Manipulation of PT cell activity affected beta/gamma band power and learning. Our results suggest that PT cell-dependent oscillations play important roles in motor learning.

[1] Division of Cerebral Circuitry, National Institute for Physiological Sciences, Okazaki, Japan. [2] Department of Physiological Sciences, Graduate University for Advanced Studies (SOKENDAI), Okazaki, Japan. [3] Brain Science Institute, Tamagawa University, Machida, Japan. [4] Present address: Section of Cellular Electrophysiology, National Institute for Physiological Sciences, Okazaki, Japan. ✉email: otsuka@nips.ac.jp

The neocortex is composed of multiple types of excitatory principal cells and inhibitory interneurons, which form complicated networks[1–3]. Information is processed through inter- and intra-laminar connections in the cortex and routed to several subcortical and other cortical areas through different sets of pyramidal cells (PCs). It is a fundamental question of how the cortex processes information and regulates high-order brain functions. Recent studies have revealed that layer V (L5) PCs, the main output cells of the motor cortex, have distinct morphological and physiological properties[4–7]. Moreover, intra- and inter-laminar synaptic connections between PCs depend on PC projection subtypes[4,6,8–10]. These observations suggest that cortical PC networks are segregated into functional channels corresponding to different target areas. Segregation of information processing within the cortical circuitry implies that individual PC subtypes play distinct functional roles in brain function. Recent studies have suggested the contribution of specific PC subtypes in learned behavior[11,12]. However, it remains unknown how individual PC subtypes regulate cortical network activity and, thereby, brain function.

Among the characteristic network activity generated in the brain, the cortex induces oscillations that include beta (15–30 Hz) and gamma (30–90 Hz) waves. These oscillations increase the power during working memory or attention[13–16]. Artificial stimulation of the cortex at these frequencies promotes behavioral learning[17,18], suggesting crucial roles of cortical oscillations in mechanisms underlying learning. Moreover, it has been proposed that beta and gamma oscillations regulate information transfer between cortical areas[19,20]. To understand mechanisms underlying oscillations and relevant functional roles, it is necessary to identify the neurons that generate oscillations and their neural connections, and elucidate the relationship between these oscillation-evoked circuits and behavioral learning. In the present study, we investigated cortical network activity induced by optogenetic manipulation of activity in specific PC subtypes in the motor cortex. We found that oscillatory activity, ranging in the high beta/low gamma band (~30 Hz), was induced in L5 PCs, as well as in fast-spiking (FS) interneurons, and was dependent on the PC subtype. Our experimental and simulation results suggest that oscillatory activity is internally generated within L5 local circuits. Oscillations at a similar frequency band were also observed in local field potentials (LFPs) recorded in the motor cortex during motor pattern learning. Optogenetic manipulation of PT cell activity simultaneously affected the power of the beta/gamma frequency band and learning ability. Our results suggest that oscillatory activity is generated through reciprocal interactions between PT and FS cells and regulates motor learning.

## Results

To understand how L5 cell activity is regulated during cortical information processing, we examined response patterns of L5 cells evoked by L2/3 PC stimulation, a major feed-forward excitatory pathway in the motor cortex. Channelrhodopsin-2 (ChR2) was selectively expressed in L2/3 PCs in the rostral part of the cortex including motor areas, using in utero electroporation at E17.5 (Fig. 1a, b). L2/3 PCs expressing ChR2 were depolarized with a short duration of blue light illumination, depending on light intensity (Fig. 1c). Light illumination of $0.25 \pm 0.05$ mW intensity (duration, 5 ms) was sufficient to induce a spike ($n = 10$ cells), indicating functional expression of ChR2 in electroporated cells. To examine network activity in the motor cortex, we delivered ramp-shaped light illumination, where light intensity was gradually increased, to L2/3 in slice preparations. During ramp-shaped light illumination, oscillatory membrane activity was observed in L5 PCs, as well as in FS cells (Fig. 1d–f). The peak frequency of induced oscillations was $30.9 \pm 3.5$ Hz in PCs and $31.5 \pm 3.3$ Hz in FS cells (Fig. 1f insets). These results indicate that the motor cortex can generate high beta/low gamma band (beta/gamma) oscillations. Induction of oscillatory activity by ramp-shaped photostimulation to L2/3 PCs expressing ChR2 was also reported in the somatosensory and visual cortices[21].

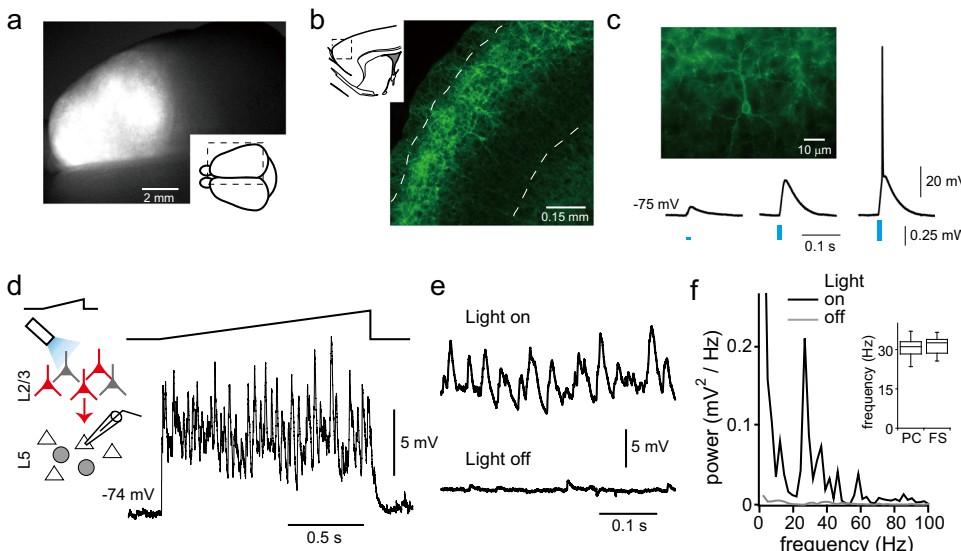

**Fig. 1 Photostimulation-induced oscillations in L5 cells. a** Low-magnification view of a rat brain expressing mCherry and ChR2-Venus by in utero electroporation at E17.5. **b** Sagittal section of frontal cortex. ChR2-Venus was selectively expressed in L2/3 PCs. Dashed lines indicate the boarder of L1/L2, and L2/3/L5. Inset, schematic view of whole (**a**) and sagittal section (**b**) of the brain; dashed rectangles correspond to the images. **c** Responses to brief light illumination (5 ms in duration; 3 intensities) in a ChR2 positive L2/3 PC. **d** Oscillatory membrane potentials of a L5 PC in response to ramp-shaped light stimulation to L2/3. Inset, schematic drawing of experiment in the slice preparation. Triangle and gray circle indicate L5 PCs and FS cells. respectively. Red and gray cells in L2/3 indicate ChR2 positive and negative L2/3 PCs, respectively. Ramp shape line indicates intensity change of light stimulation. **e** Magnified trace of membrane potentials of the cell shown in (**d**) during stimulation (upper) and no stimulation (lower). **f** Power spectra of membrane potentials. Inset, box plots of peak frequencies around 25–45 Hz induced by light stimulation in L5 PCs and FS cells ($n = 60$ and 42, respectively).

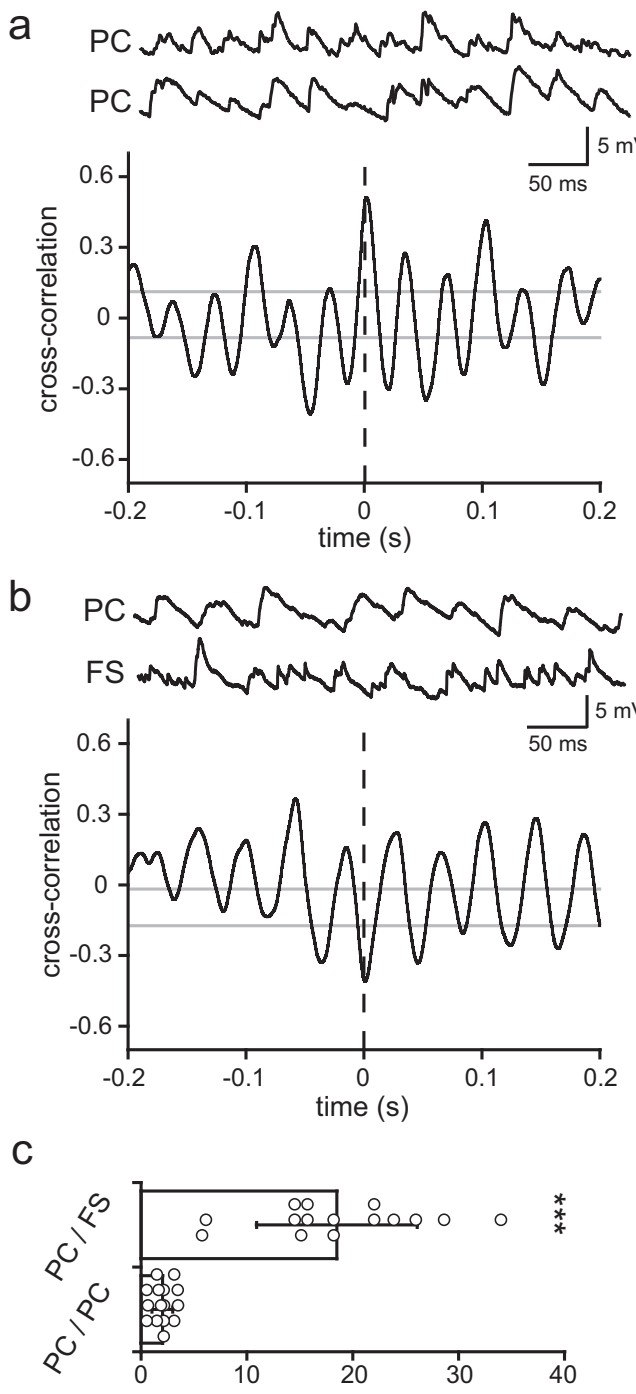

**Fig. 2 Cross-correlation analysis between L5 cells during oscillatory activity. a** Dual recordings from L5 PCs (upper) and a cross-correlogram. **b** Dual recordings from a PC and a FS cell in L5, and a cross-correlogram (PC, point of reference). Dashed line indicates time 0. Gray lines indicate 95% confident intervals. **c** Temporal differences between time 0 and the nearest positive peak in cross-correlograms of PC/PC pairs ($n = 15$) and PC/FS cell pairs ($n = 15$). ***, a significant difference ($P < 0.0001$, Wilcoxon signed-rank test). Data are expressed as mean ± SD.

To understand how oscillatory activity is generated in cortical circuits, we performed dual recordings from L5 cells and examined the temporal relation of membrane potentials during oscillation induced by light stimulations to L2/3 PCs. Without light stimulation, L5 PC and FS cell received spontaneous excitatory

synaptic currents (sEPSCs) at $3.02 \pm 1.18$ and $12.83 \pm 2.2$ Hz, respectively. Synchronized sEPSCs between two cells, which occurred within 5 ms, were rarely found in both PC/PC cell pairs ($0.05 \pm 0.02$ Hz; 4 out of 5 pairs) and PC/FS cell pairs ($0.14 \pm 0.05$ Hz; 5 pairs). During light stimulations, both PC/PC and PC/FS cell pairs showed well-synchronized membrane potential oscillations during light stimulations (Fig. 2a, b). The oscillatory phases were coherent between PC/PC pairs (nearest positive peak from time $0 = 2.09 \pm 1.04$ ms, 15 cell pairs) but were not matched between PC/FS cell pairs (nearest positive peak $= 18.43 \pm 9.08$ ms, 15 cell pairs; $P < 0.0001$, Wilcoxon signed-rank test, Fig. 2c). Oscillatory activity during light stimulation was also observed in L5 non-FS interneurons ($29.21 \pm 3.49$ Hz in peak frequency, $n = 15$), but the correlation with PCs was weak (Supplementary Fig. 1). Taken together, these results suggest that L5 PCs and FS cells are depolarized at different phases of the beta/gamma oscillations induced by L2/3 PC stimulation. If oscillatory activity generated by L2/3 was simply transmitted to L5 cells, both PCs and FS cells would be synchronously depolarized. Furthermore, L5 PCs and FS cells frequently form reciprocal connections between them, but receive common excitatory inputs from L2/3 PCs with a very low probability[22], suggesting generation of oscillatory activity at L5 local circuits, using reciprocal connections between L5 PCs and FS cells.

**Oscillatory activity in PC subtypes.** L5 PCs are largely categorized into two types, pyramidal tract (PT) and intratelencephalically projecting (IT) cells[23,24]. PT and IT cells form synaptic connections among them in a different manner[4,6–9]. We examined how L5 PT and IT cells are involved in the generation of oscillatory activity. PT and IT type PCs were simultaneously identified by fluorescent retrograde tracer injections to the target areas (Fig. 3a). Dual recordings were obtained from L5 PT and IT cells, while ramp-shaped light illumination was applied to L2/3. Membrane potential oscillations during photostimulations were frequently observed in PT cells, but not in the majority of IT cells (Fig. 3b, c). In addition, hyperpolarization of membrane potentials during stimulation was often observed in IT cells (11 out of 27 cells, peak amplitude $= -1.74 \pm 0.37$ mV, voltage area $= 0.73 \pm 0.16$ mV*s, Fig. 3b). These results suggest that PT cells are involved in the generation of oscillatory activity.

It has been proposed that reciprocal synaptic interaction between excitatory PCs and inhibitory cells is one of the major mechanisms underlying the generation of oscillatory activity[25]. The above observations taken together with this proposal suggest that L5 PT and FS cells, rather than IT and non-FS cells, discharge spikes during L2/3 photostimulation. To confirm this, spiking activity was recorded from L5 PC subtypes in cell-attached mode to preserve the intracellular circumstances of cells. In most recorded IT cells, no spikes were observed during L2/3 photostimulation (11 out of 12 cells). By contrast, 11 out of 17 PT cells generated spikes in response to L2/3 photostimulation (Fig. 3d, e). The number of spikes was $5.2 \pm 5.04$ in PT cells that exhibited at least one spike during stimulation (duration, 2 s). In spike recordings from inhibitory interneurons, identified cell types based on firing properties in whole-cell mode following cell-attached recordings, L5 FS cells were found to discharge more spikes during the photostimulation period than were non-FS cells (Fig. 3d, e). These results suggest that synaptic interactions between L5 PT and FS cells generate oscillatory activity in L5 cortical circuits.

Electrical connections between FS cells via gap junctions play important roles in regulating FS cell activity[26]. Electrically connected FS cells frequently receive common inputs from surrounding PCs, which could generate a synchronous activity between them[27]. We, therefore, examined whether electrical

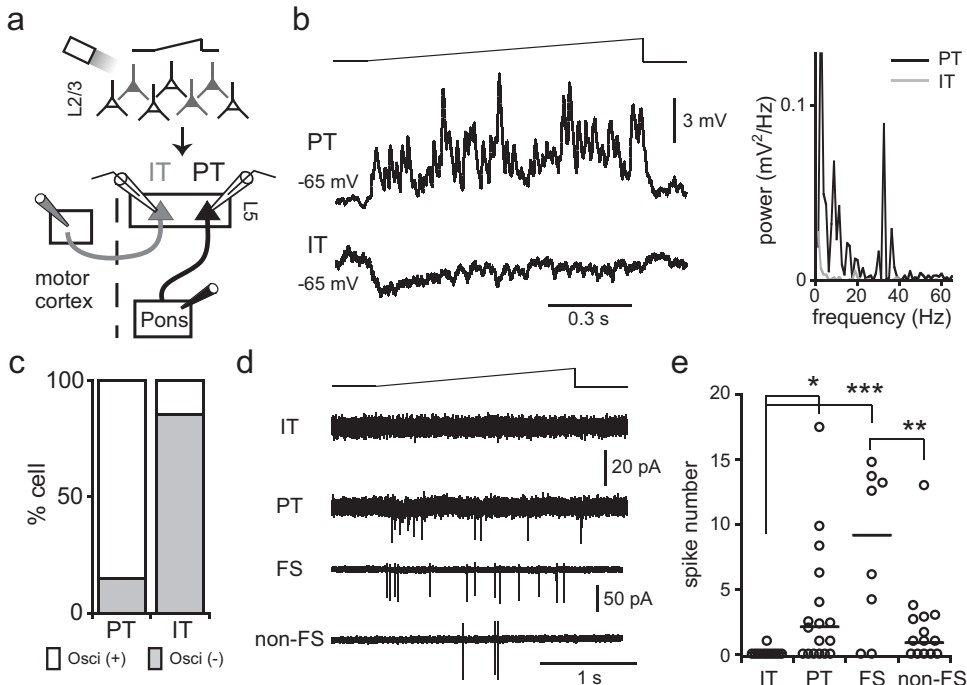

**Fig. 3 Responses to L2/3 PC photostimulation in L5 cell subtypes. a** Identification of L5 PC subtypes by retrograde fluorescent tracer injections to contralateral cortex and ipsilateral pontine nuclei. **b** Dual recordings from identified PT and IT cells. Ramp-shaped light stimulation to L2/3 induced oscillations in PT cells, but not IT cells. Right graph, power spectra of membrane potentials of the PT and IT cells during stimulation. **c** Proportion of PT and IT cells that exhibited oscillatory activity, respectively (27 PT/IT cell-pairs). Oscillations were adopted when the ratio of peak power (25–40 Hz band) to averaged power (45–55 Hz band) >10, and peak power (25–40 Hz band) >0.5 mV$^2$ Hz$^{-1}$. **d** Cell-attached recordings obtained from IT, PT, FS, and non-FS cells during ramp-shaped stimulation to L2/3 (upper trace). **e** Number of spikes induced in individual cell types during photostimulation. The horizontal bar indicates median number of spikes. Asterisk indicates significant difference (one-way ANOVA, $F_{(3, 49)} = 8.202$, $P = 0.00016$; Bonferroni test, IT vs. PT, $P = 0.0359$, IT vs. FS, $P = 0.0001$, FS vs. non-FS, $P = 0.0071$).

connections between FS cells were necessary for L5 beta/gamma oscillations induced by L2/3 PC stimulation. A gap junction blocker suppressed oscillatory activity in both PCs and FS cells (Supplementary Fig. 2). These results suggest that electrical connections between FS cells are involved in the generation of beta/gamma oscillations in cortical circuits.

**Oscillations induced by L5 PC subtype stimulation.** We examined whether activation of specific L5 PC subtypes is sufficient to induce beta/gamma oscillations in L5. The cortex develops from deep to superficial layers in an inside-out manner. Thus, either L2/3 or L5 PCs can be selectively targeted using in utero electroporation performed at different embryonic days[28]. Moreover, L5 PT and IT cells are generated at different times in the mouse[29]. We performed in utero electroporation at E14.5 or E15.5 and examined the expression of CTIP2, a marker for PT cells[30,31], in labeled L5 PCs. In rats electroporated with mCherry at E15.5, only 2.82 ± 1.02% of labeled L5 cells were CTIP2 positive. By contrast, most labeled L5 cells were CTIP2 positive in rats electroporated at E14.5 (96.92 ± 1.05%, six animals, Fig. 4a, b). In both cases, labeled cells were mostly positioned within L5 (Supplementary Fig. 3). Consistent with the distribution of L5 PT and IT cells[32], the distribution of labeled cells within L5 were somewhat different between rats electroporated at E15.5 and E14.5. Labeled cells at E15.5 were located in the upper L5 more than those in lower L5. By contrast, the proportion of labeled cells at E14.5 located in lower L5 were increased. These results indicate that PT and IT cells can be selectively targeted by in utero electroporation at different embryonic days.

To examine the activity induced by specific L5 PC subtype stimulation, recordings were obtained from ChR2-negative L5

PCs and FS cells (Fig. 4c). Brief photostimulation (duration, 5 ms) of L5 induced transient synaptic responses in L5 cells of rats electroporated at E15.5 or E14.5 (data not shown). In response to ramp-shaped light illumination of L5, oscillatory activity was observed in some L5 PCs and FS cells of rats electroporated at E14.5 (PT stimulation; Fig. 4e; peak frequency = 31.1 ± 3.05 Hz) and in few cells of rats electroporated at E15.5 (IT stimulation; Fig. 4d). Moreover, oscillatory activity induced by PT stimulation was highly dependent on PC subtype. We classified PCs based on firing properties, because it is difficult to identify L5 PC subtypes by retrograde tracers in animals with fluorescently labeled L5 PCs by in utero electroporation. Firing properties in L5 PCs have been shown to correlate with their projection areas[4–6]. PT cells show repetitive spikes during current pulse injections with slow spike frequency adaptation (SA type). In particular, SA type cells with initial doublet spike burst (SA-d) have only been observed in PT cells[4]. On the contrary, the majority of IT cells show fast spike frequency adaptation in response to current pulse injections (FA type). Utilizing identification based on firing properties of L5 PC subtypes, we compared their response patterns to ramp-shaped light illuminations. Oscillatory activity induced by PT cell stimulation was found in 15 out of 19 L5 PT type (SA-d) and in 2 out of 16 IT type (FA) PCs (Fig. 4f). These results suggest that PT cells are involved in the generation of oscillatory activity within the local circuitry in L5.

**Oscillatory activity in circuit model.** Our experimental data suggest that interactions between PT and FS cell subnetworks play a key role in the generation of L5 oscillations. Previous studies have reported unique characteristics of PT cell subnetwork in connection patterns, synaptic properties, and firing

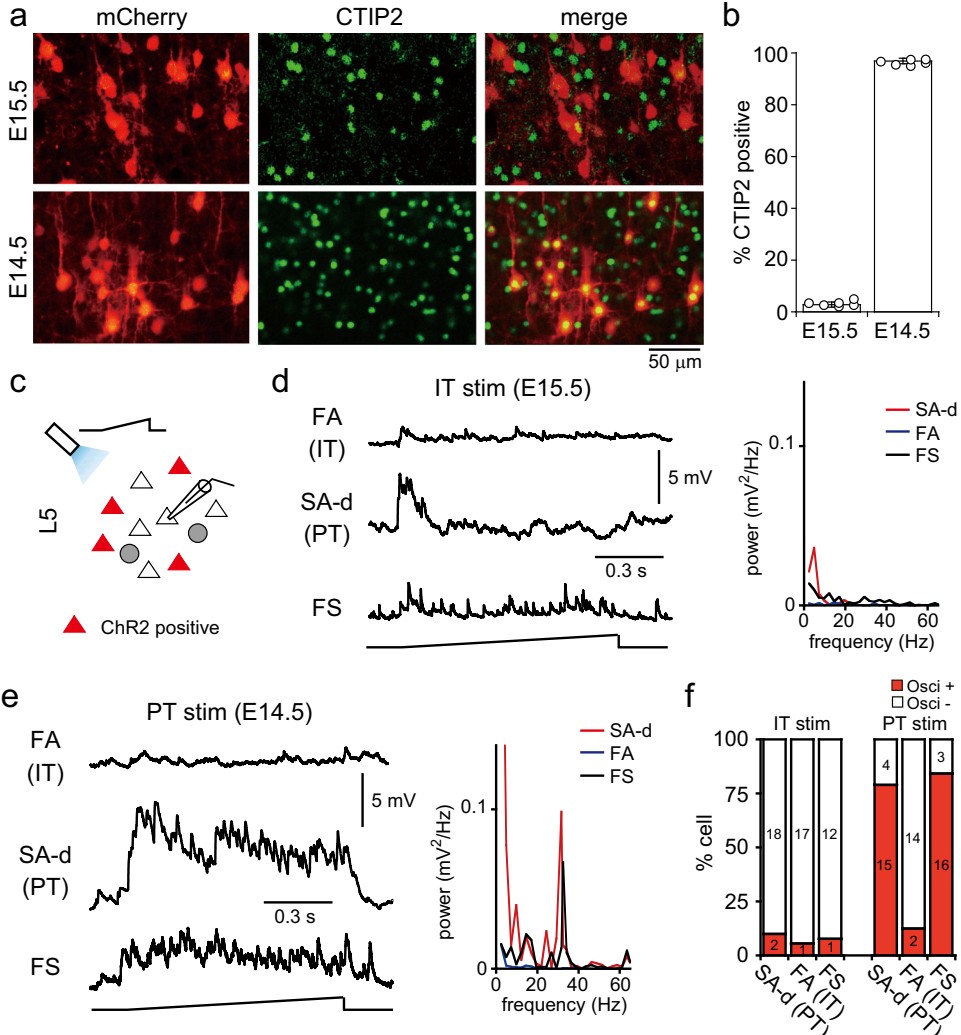

**Fig. 4 Oscillations induced by specific PC subtype stimulation. a** CTIP2 expression in L5 PCs transfected with mCherry by electroporation at E15.5 or E14.5. **b** Proportion of CTIP2-positive cells among PCs expressing mCherry (six rats, respectively). Data are expressed as mean ± SD. **c** Schema of L5 photostimulation and recording from non-labeled cells. **d** Recordings from L5 cell-subtypes during ramp-shaped photostimulation to L5 in slices obtained from rats electroporated with ChR2 at E15.5 (stimulation of IT cell population). PC firing subtypes: SA-d (slow spiking adaptation with initial doublet spikes) type, overlapped considerably with PT cell population; FA (fast-spiking adaptation) type, overlapped with IT cell population. Right graph, power spectra of membrane potentials during stimulation. **e** Membrane potentials of L5 subtypes during photostimulation in slices obtained from rats electroporated with ChR2 at E14.5 (stimulation of PT cell population). Right, power spectra. **f** Proportion of L5 cells exhibiting oscillations upon photostimulation of IT or PT cell populations. Oscillations were adopted when the ratio of peak power (25–40 Hz band) to averaged power (45–55 Hz band) >10, and peak power (25–40 Hz band) >0.5 mV$^2$ Hz$^{-1}$.

properties of PT cells in the rat frontal cortex[4,7,8,22]. PT cells discharge spikes with slow spike adaptation, in response to depolarization, and form synaptic connections that show connection reciprocity and short-term facilitation in response to repetitive spikes. These properties might underlie the generation of oscillatory activity in L5. To confirm this notion, we constructed L5 model circuits based on experimental data. Our model consisted of 400 PCs (200 PT and IT type cells, respectively) and 50 FS cells distributed randomly on a 500- μm-sided square plane. PT, IT, and FS model cells reproduced the firing properties observed in the experiments (Fig. 5a). Synaptic connections including electrical synapses between FS cells were made between model cells located within a 150-μm distance of each other at connection probabilities obtained from slice experiments (Fig. 5b). To induce network activity in the model circuits, external synaptic inputs were randomly applied to individual cells at input frequencies reflecting relative connection probabilities

from L2/3 to L5 cell types[4,22]. In the presence of external inputs in the model, oscillations were evoked in PT and FS model cells at frequencies similar to those observed in slice experiments (Fig. 5c). These results suggest that L5 PT and FS cell subnetworks act as oscillation generators.

We identified critical factors for the generation of oscillations in the model. When firing properties of PT cells were switched to those of the IT type that showed fast spike adaptation during depolarization, oscillations were suppressed (Fig. 5d). We also examined the contribution of connectional features between PT cells to the generation of oscillations. Alteration of synaptic properties between PT cells to the depression type abolished oscillations (Fig. 5e, left). Reduction in reciprocal connection probability between PT cells disrupted the generation of oscillatory activity (Fig. 5e middle). In accordance with experimental observations, the elimination of electrical synapses between FS cells suppressed the oscillatory activity (Fig. 5e,

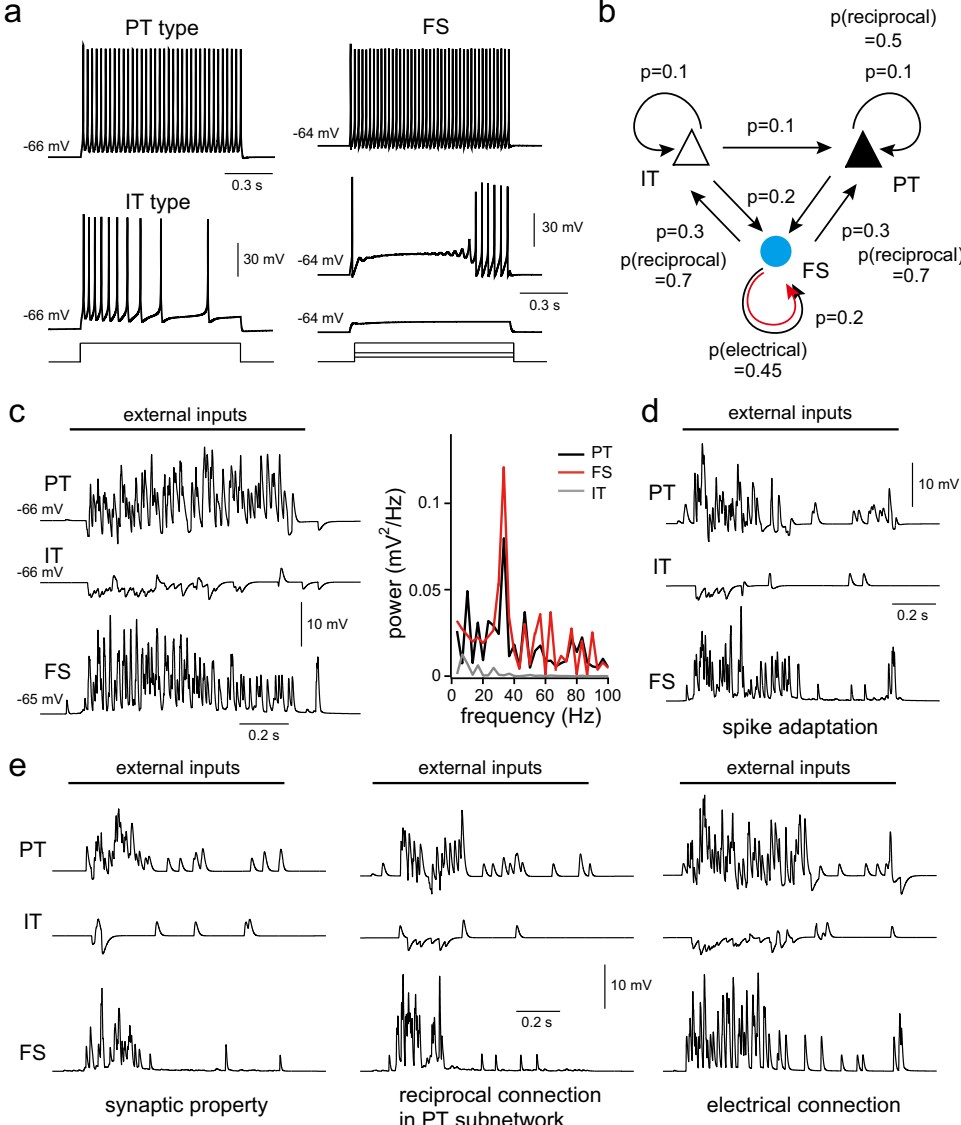

**Fig. 5 Induction of oscillations in L5 model circuits. a** Firing properties of PT, IT, and FS cell models in response to somatic current injection.
**b** Connectional features of L5 cells in the model circuits. Black and red arrow indicates chemical and electrical synaptic connection, respectively. 'p'
indicates synaptic connection probability to surrounding cells located within a 150 μm distance. 'p(reciprocal)' indicates reciprocal connection probability
among connected cells. **c** Somatic membrane potentials of PT, IT, and FS cells in the network model. External inputs were applied to all cell models.
Right graph indicates power spectra of membrane potentials in the presence of external inputs. **c**, **e** Somatic membrane potentials of PT, IT, FS cells in the
model with modification of model parameters. **d** Firing properties in PT cell models were changed from slow to fast spike adaptation type by increasing
the M type potassium conductance to 1. **e** Manipulation of connection properties in PT and FS cell subnetworks in the model. Left, short-term plasticity of
PT cells was changed from facilitation to depression type, similar to that of IT cells. Middle, reduction of the reciprocal connection probability in the PT
cell subnetwork from 0.5 to 0.1. Right, removal of electrical connections among FS cells. Sodium conductance was set to 0 in cells shown in (**c–e**).

right). These results demonstrate that L5 circuits are well organized to generate oscillations within specific excitatory and inhibitory subnetworks.

**Oscillatory activity during motor learning**. Oscillatory activity in the motor cortex depends on behavioral states[33–36]. We investigated what kinds of behavioral conditions induced beta/gamma oscillations observed in in vitro experiments to understand their functional roles in the motor cortex. We compared LFPs in L5 of the motor cortex that occurred during simple locomotive movements with that observed during more difficult movements that need to be learned. We adopted a forced wheel running task, in which animals ran on foot bars placed along a

rotating wheel rim in order to drink water from the supply port[37]. Rats were habituated to running on regular-interval bars (regular task), followed by running on patterned-interval bars (pattern learning task) (Fig. 6a, b). During the initial stage of learning trials, animals frequently fell down from the step bars and touched the foot floor. However, the number of touches on the floor gradually decreased across trials, indicating learning of the bar pattern (Fig. 6c, supplementary Fig. 4, and movie 1). We next obtained LFPs from the M1 forelimb area at rest (no wheel rotation), during regular tasks, and during pattern learning tasks at the initial learning stage (day 1) (Fig. 6d). In the power spectra, we found three frequency bands that increased during tasks: 5–10 Hz (theta band) and 60–70 Hz during regular and pattern learning tasks; and 25–45 Hz (beta/gamma frequency band) only

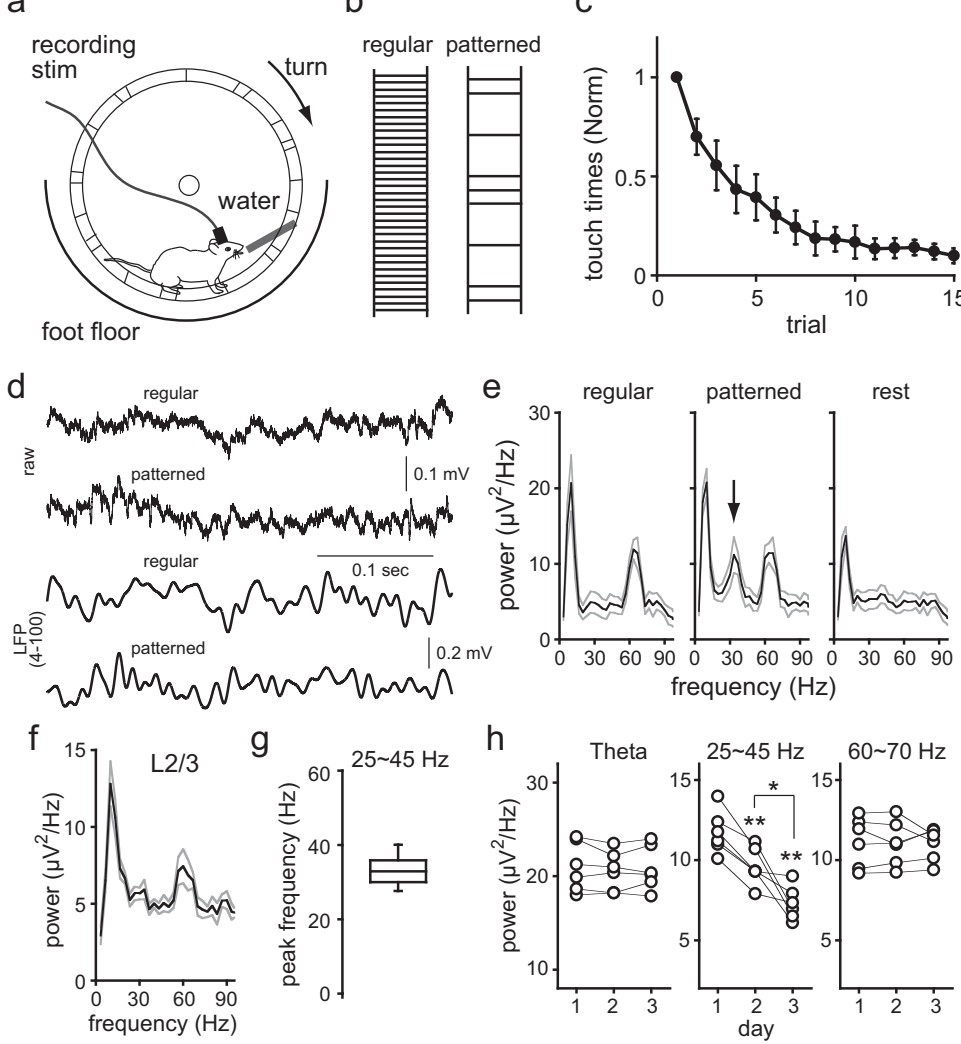

**Fig. 6 Motor learning accompanied by LFP alterations. a** Schematic drawing of the wheel running task. The wheel was driven by a motor at constant speed. Water supply port was inserted from the side of wheel. **b** Foot bar patterns used for habituation (regular task) and for learning (pattern learning task). **c** Learning progress monitored by reduction in the number of forelimb touches with the floor during the learning task from day 1 to day 3 (5 trials per day). Number of touches was normalized to that of the first trial. Data are expressed as mean ± SD (n=18 rats implanted optical cannulas). **d** Raw traces (upper) and LFPs (lower) during the regular task and pattern learning task (day 1). The LFPs were processed by a band-pass filter (4–100 Hz). **e** Power spectra of LFPs at rest ($n = 12$), during the regular task ($n = 15$), and during pattern learning task ($n = 18$) obtained on day 1. The power increased around 30–40 Hz in the pattern learning task (arrow), but was not changed around theta and 60–70 Hz in both the regular and learning tasks. Power spectra were represented as mean ± SEM (black and gray line, respectively). **f** Power spectra of LFPs, represented as mean ± SEM (black and gray line, respectively), obtained at L2/3 during the learning task on day 1 ($n = 10$). **g** Box plot of peak frequencies between 25 and 45 Hz in the LFP power spectra in L5 during the pattern learning task (day 1; $n = 18$). **h** Power at a peak frequency of theta, 25–45, and 60–70 Hz frequency band of LFPs obtained on days 1–3 of the learning task. Note significant decrease in power around 25–45 Hz through the trial days (paired $t$-test, day 1 vs day2, **$P = 0.00038$, day2 vs day 3, *$P = 0.0124$, day 1 vs day 3, **$P = 0.0012$).

during pattern learning tasks (Fig. 6e). The peak frequency of the beta/gamma band in L5 was $32.9 \pm 3.4$ Hz (Fig. 6g, 18 rats), similar to that observed in PCs and FS cells in slice experiments (one-way ANOVA, $F_{(2, 117)}=2.415$, $P = 0.0938$; Bonferroni test, versus PCs, $P = 0.0937$, versus FS cells, $P = 0.4821$). The increase of beta/gamma frequency band was layer specific. In LFPs obtained in L2/3 at day 1, beta/gamma-band activity was weak during tasks (Fig. 6f). In the current source density analysis (CSD), the current sink and source components were observed within L5 (Supplementary Fig. 5). These results suggest that beta/gamma-band activity is generated in L5 local circuits.

During a learning trial, wheel rotation speed was changed to examine whether locomotion activity affects these frequency bands. Consistent with the previous study[38], theta and 60–70 Hz

frequency bands were changed, depending on the speed of locomotion. On the other hand, beta/gamma frequency band was not changed (Supplementary Fig. 6). These results suggest that theta and 60–70 Hz frequency bands are related to locomotion itself, but that of beta/gamma band is related to the learning process. Indeed, the power of beta/gamma-band gradually decreased through trial days (Fig. 6h). Theta and 60–70 Hz frequency band activity remained constant.

If beta/gamma-band activity is necessary for motor learning, manipulation of this activity during tasks would affect the learning process. To this end, we examined the effects of photostimulation on LFPs by ramp-shaped light illumination, similar to those in slice experiments, to the M1 forelimb area in rats expressing ChR2 in L2/3 PCs. Photostimulation increased the

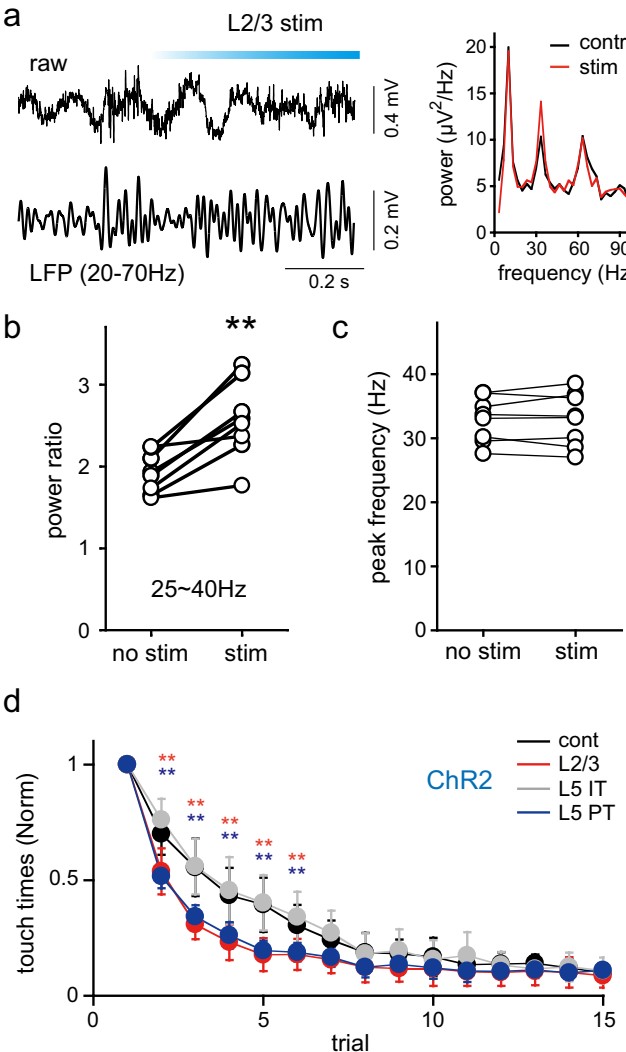

**Fig. 7 Effects of selective stimulation of PC subtypes on oscillations and motor learning. a** Raw trace (upper) and LFPs (lower), band-pass filtered between 20 and 70 Hz, observed during the pattern learning task on day 1. L2/3 cells were stimulated by ramp-shaped light illumination. Right, power spectra of LFPs recorded in the presence and absence of light illumination. **b** Ratios of power at peak frequency between 25 and 45 Hz to averaged power between 45 and 55 Hz in the presence and absence of photostimulation (**, Wilcoxon signed-rank test, *P* = 0.0078, 8 rats). **c** Peak frequencies of LFP power spectra between 25 and 45 Hz obtained during a learning task in the presence and absence of photostimulation (8 rats). **d** Effects of ChR2 photostimulation on learning progress. Number of touches by forelimbs to the foot floor normalized to that of the first trial. Photostimulation was applied to both hemispheres of rats expressing ChR2 by electroporation at E17.5 (L2/3 PC, red, *n* = 12), E15.5 (IT, gray, *n* = 10), or E14.5 (PT, blue, *n* = 11). Rats implanted with cannulas and connected with an optical fiber during trials, but without stimulation, were used as controls (black, *n* = 18). Note that learning was significantly accelerated by L2/3 and L5 PT cell stimulation. **P < 0.001 in red for L2/3 vs. control and vs. IT, and in blue for PT vs. control and vs. IT (Bonferroni post hoc test). *P* values are shown in Supplementary Table 2. Data are expressed as means ± SD.

power of beta/gamma-band selectively (34.67 ± 0.16%, Wilcoxon signed-rank test, *P* = 0.0078, 8 rats, Fig. 7a, b). However, the peak frequency of this band did not shift during photostimulation (Fig. 7c). By contrast, the power of 60–70 Hz band did not change during stimulation (peak power and frequency, −1.46 ± 9.37% and −1.7 ± 4.2% change, respectively). Theta band activity was

also preserved during stimulation (peak power and frequency, −0.07 ± 2.8% and 0.75 ± 14.32% change, respectively).

In parallel with the stimulated increase in beta/gamma power, learning proceeded faster with L2/3 PC photostimulation in rats expressing ChR2 in L2/3 PCs than those without it (Fig. 7d). Moreover, consistent with observations in slice experiments, PT, but not IT, cell stimulation accelerated learning (Fig. 7d). Stimulation of PCs may affect locomotive behavior itself rather than motor learning. We therefore compared moving distances among rats with selective photostimulation of individual PC subtypes in an open field test. However, no significant differences were found between control, L2/3, PT, and IT cell-stimulated animals (Supplementary Fig. 7), suggesting that the optogenetic stimulation used in the learning task did not affect locomotion.

Either L2/3 or L5 PT type PC activation by light illumination accelerated motor learning. To further determine the involvement of specific PC subtypes in learning, we selectively suppressed individual PC subtype activity during learning tasks, using continuous Archaerhodopsin (eArch) activation (Supplementary Fig. 8). In contrast to ChR2 activation, time courses of learning were significantly slowed down in rats expressing eArch in L5 PT cells, but not in those expressing eArch in L2/3 or L5 IT type PCs (Fig. 8a). We examined whether beta/gamma-band power was affected by selective suppression of PC subtype activity. LFPs during photostimulation were recorded during pattern learning tasks on day 1. The power of the beta/gamma band in LFPs during a task was inhibited when L5 PT, but not L2/3 or L5 IT, cell activity was suppressed (Fig. 8b, c). The power of 60–70 Hz band did not change during suppression (% change of peak power and peak frequency; 0.41 ± 3.57 and −1.36 ± 2.66 in PT cells, 0.66 ± 3.45 and −0.43 ± 2.36 in IT cells, and 2.03 ± 2.92 and −0.11 ± 3.04 in L2/3 PCs). Theta frequency band activity was also preserved during stimulation (% change of peak power and peak frequency; 0.41 ± 1.95 and 5.93 ± 6.65 in PT cells, 0.72 ± 1.44 and −1.68 ± 8.14 in IT cells, and 0.92 ± 2.3 and 1.37 ± 8.71 in L2/3 PCs). We also confirmed locomotion was not affected by eArch suppression of L5 PT, L5 IT, or L2/3 PCs using the open field test (Supplementary Fig. 7). Taken together, these results suggest that PT cells play important roles in the generation of oscillatory activity related to motor learning.

## Discussion

In the present study, we investigated how PC subtypes contribute to cortical network activity. We found that oscillations ranging in the beta/gamma frequency band were induced in vitro in L5 PT cells as well as in FS cells. Oscillations in a similar frequency band were also observed in LFPs in vivo recorded during a motor learning task. Optogenetic manipulation of PT cell activity simultaneously affected oscillatory activity and motor learning. Our results suggest that beta/gamma oscillations are locally generated by reciprocal interactions between PT and FS cell subnetworks and play important roles in the progression of the learning process (Fig. 8d).

A previous study reported that excitation of L2/3 PCs can induce oscillatory activity in the somatosensory and visual cortices[21]. Consistent with this, we found that, in the motor cortex, excitation of a subset of L2/3 PCs induced oscillatory activity in L5 cells. Moreover, we found that the L5 local circuit itself can generate beta/gamma oscillations that are dependent on L5 PT cells. This is also supported by the observation that PT cell spiking activity is phase-locked to 20–30 Hz oscillations in the motor cortex during a task[39]. Cortical PCs form subnetworks in intra- and inter-laminar connections, depending on projection targets. PT cells exhibit sustained spiking activity during excitation and frequently form reciprocal connections with each other,

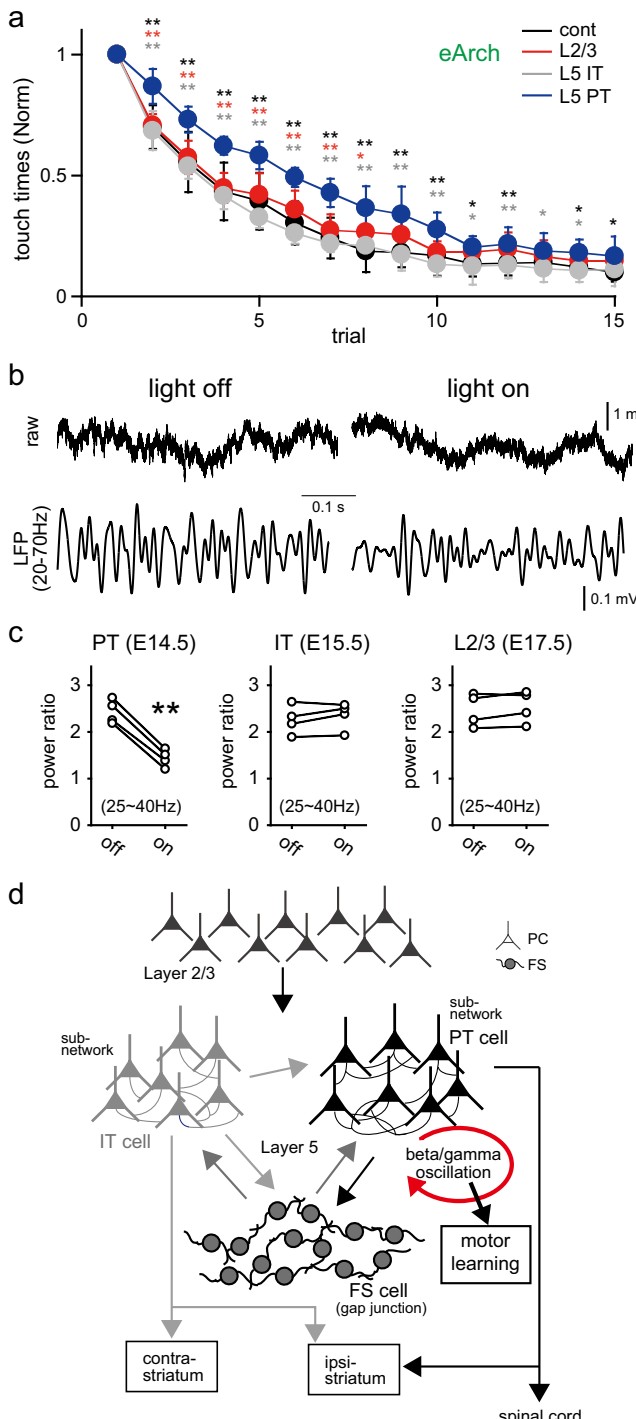

**Fig. 8 Effects of selective suppression of PC subtypes on motor learning and oscillations. a** Pattern learning task using rats expressing eArch in L2/3 PCs (red, n=11), L5 IT cells (gray, n = 10), or L5 PT cells (blue, n = 11) of both hemispheres by electroporation at E17.5, E15.5, or E14.5. During task, 1 mW green light was continuously applied to both sides. Number of floor touches of forelimbs was normalized to that of the first trial. Control rats (black, n = 18) were the same as in ChR2 stimulation experiments. Note that learning was significantly slowed down by L5 PT cell inhibition. (**P < 0.01, *P < 0.05 in black, red, and gray for PT vs. control, PT vs. L2/3, and PT vs. IT, respectively). P values are shown in Supplementary Table 3. Data are expressed as means ± SD. **b** Raw traces (upper) and LFPs (lower) recorded in the presence and absence of light illumination to animals expressing eArch in L5 PT cells during a learning task on day 1. **c** Ratios of power at a peak frequency between 25 and 45 Hz to averaged power between 45 and 55 Hz during learning tasks. Power at peak frequency was significantly suppressed by L5 PT cell inhibition (four rats in individual group; **paired t-test, F = 3, P = 0.00023). **d** Schematic drawing of motor cortical circuitry. L5 PCs (black, PT cell; gray, IT cell) form excitatory subnetworks depending on their projection subtypes and reciprocally interact with inhibitory FS interneurons (filled circle) that also form subnetworks via dendritic gap junctions. Beta/gamma oscillations are generated by interactions between PT and FS cell subnetworks (red arrow) and regulates motor learning. Arrows between cell types indicate the direction of synaptic connections.

inhibitory connections that are stronger than unreciprocated connections[22,42,43]. In addition, excitatory inputs from PT cells to FS cells show weaker short-term depression than those from IT cells[43]. These connection patterns suggest close interactions between PCs, especially PT cells, and FS cells[44]. Moreover, FS cells selectively make electrical connections among themselves and form dendritic net structures in the cortex[27,45–47]. Electrically connected FS cells frequently receive common inputs from surrounding PCs[27]. These connection patterns could enhance synchronized activity between FS cells and contribute to the generation of oscillations in the network. However, the involvement of electrical synapses in the generation of oscillatory activity is still controversial[48–50].

We detected a distinct beta/gamma band during the learning progress period. Beta oscillations previously investigated in the motor cortex were not found to be induced during actual limb movements, but intensified during holding a position or while directing attention toward following a task before performing an action, and potentiated after incorrect executions[33,34,36,51]. However, based on our findings that learning progress correlated with beta/gamma power occurring either naturally or when modulated by optogenetic manipulation, we propose that acquisition of skilled behaviors needs the increase in beta/gamma-band power, generated locally by L5 PT and FS cell interactions in the motor cortex. After learning, beta/gamma activity would be not necessary to execute acquired behavior, since beta/gamma power decreased as learning progressed. The oscillatory activity would enhance synchronous activity within the network and oscillatory coupling with the projection areas that induce plasticity for learning[52]. Manipulation of oscillatory activity power would affect learning progress by changing synchronization and coupling strength with the projection area. Indeed, optogenetic manipulation of beta/gamma-band activity affected time courses of learning progress in the present study.

In rodents, theta frequency band activity is modulated by motor learning[53,54]. Increase of theta power accompanies phase-locked spiking activity in the motor cortex as well as the striatum. These would play important roles in precise movements of learned skills[53]. In our study, theta frequency band activity was affected by movement speed rather than learning. How different

which are characterized by a facilitation property in response to repetitive spiking activity[4,7,8]. The requirements of these properties were confirmed in silico by manipulation of relevant parameters in our computational model. During development, L5 cells cannot induce oscillation by optogenetic stimulations[40], suggesting that maturation of network and cell properties is necessary to induce oscillatory activity.

It has been proposed that recurrent connections between PCs and FS cells underlie the generation of oscillations in the cortex and hippocampus[25,41]. Consistent with this notion, our experimental and simulation studies suggest that recurrent connections between PT and FS cells generate oscillations in cortical circuits. Cortical PCs and FS cells frequently form reciprocal excitatory/

frequency band oscillations are involved in the learning remains unknown. In addition, another gamma band (~60 Hz) was also modulated by running, but not related to motor learning. A similar band increase has been reported in M1 during movements[38,55]. These suggest that gamma band of about 60 Hz in the motor cortex has prokinetic effects[56]. Theta and gamma band was not affected by local optogenetic manipulations of PC subtypes, and may be transmitted from other brain regions such as other cortical areas.

Although our results suggest the regulation of pattern learning by PT cells, the functional roles of IT cells in the motor cortex are still unclear. A recent study reported the importance of IT cells in the prefrontal cortex for learning of goal-directed behavior[11]. Depending on the modality of learning, different sets of PC subtypes might be involved in the learning process via distinct mechanisms. In pattern learning, c-fos expression was observed in M1, the secondary motor cortex (M2), and striatum after learning[57], suggesting the involvement of cortico-cortical and cortico-striatal pathways. M1 and M2 areas are reciprocally connected with each other[31]. M2 cells become active prior to those in M1 during a task[58]. Inputs from M2 might act as a driving force to generate oscillations in M1 during learning. Beta band oscillations are also observed in the basal ganglia including in the striatum[59,60]. This oscillation has two frequency components: low-beta (14–20 Hz) and a beta/gamma band (20–40 Hz), which are increased and weakened by dopamine depletion, respectively[61], suggesting that beta/gamma band activity in the basal ganglia regulates reward-related learning. In the motor task used here, reward motivation promoted learning. PT cells project to the subthalamic nucleus (STN) as well as to the striatum[62]. STN is a key component of the oscillation-generating network in the basal ganglia and its activity is strongly regulated by the cortex[63,64]. Thus, beta/gamma-band activity generated by the PT/FS cell network in the motor cortex may be transmitted to the striatum and/or STN and regulate motor learning.

## Methods

**In utero electroporation**. All experiments were carried out under a protocol approved by the Institutional Animal Care and Use Committee of the National Institutes for Natural Sciences. pCAGGS-ChR2-Venus[65], pCAGGS-eArch-YFP, and pCAGGS-mCherry, which are channelrhodopsin2-Venus, eArch3.0-YFP, and mCherry expression vectors, respectively, driven by the CAGGS promoter, were used. Plasmids were purified using Endofree plasmid maxi kit (Qiagen, Hamburg, Germany). Before electroporation, plasmid DNA was diluted to 1 mg ml$^{-1}$ (ChR2) or 0.5 mg ml$^{-1}$ (eArch) with 0.3–0.4 mg ml$^{-1}$ (mCherry) in phosphate buffer saline, and Fast Green solution was added to a final concentration of 0.03% to monitor injection. Pregnant Wistar rats were anesthetized with ketamine (40 mg kg$^{-1}$, i.m.) and xylazine (4 mg kg$^{-1}$, i.m.). DNA solution (~1 and 0.5 μl for E17.5 and E14.5–15.5, respectively) was injected into the lateral ventricle of embryos, using glass pipettes. To control expression sites among cortical areas, electroporation was performed with triple electrode probes[66]. The head of each embryo was held with a tweezers-type electrode (as negative poles), and a single electrode (as a positive pole) was attached to the motor cortical area. Square voltage pulses (45 and 40 V in amplitude for E17.5 and E14.5–15.5, respectively, and 50 ms in duration) were applied five times at 1 s intervals. After surgery, embryos were returned to the abdominal cavity, and the abdominal wall and skin were sutured.

**Immunohistochemistry**. Animals expressing mCherry by electroporation at E15.5 or 14.5 were perfused at 3–4 weeks of age with 3% paraformaldehyde and 0.2% picric acid in phosphate buffer. After perfusion, the brains were removed and incubated for 1 h in the fixation solution. Fixed brains were sliced into 20-μm-thick sections and incubated overnight with a rat monoclonal antibody against Ctip2 (ab18465, Abcam plc, Cambridge, UK; 1:500), a marker for PT type PCs[30], in Tris-based saline containing 10% normal goat serum, 2% bovine serum albumin, and 0.5% Triton X-100. After washes, the sections were incubated with biotinylated anti-rat IgG (1:500) and streptavidin 350 or 488 (1:250). Ctip2-positive and -negative cells were counted among cells expressing mCherry. To analyze the distribution of labeled cells by in utero electroporation, a mouse monoclonal antibody against neuronal nuclei (NeuN; MAB377, EMD Millipore Corp., Billerica, MA, USA; 1:5000) was used for layer identification.

**Slice experiments**. Brain slices were prepared from both male and female animals, expressing ChR2 and mCherry by in utero electroporation, aged postnatal (P) 21–35 days, as described previously[4]. Brains were cut in an ice-cold solution containing (in mM): 90 N-methyl-D-glucamine, 40 choline-Cl, 2 KCl, 1.25 NaH$_2$PO$_4$, 1.5 MgCl$_2$, 0.5 CaCl$_2$, 26 NaHCO$_3$, and 10 glucose (310 ± 5 mOs mL$^{-1}$, pH 7.4 adjusted with HCl). Slices including motor cortical area (300 μm thickness) were incubated in an oxygenated artificial cerebral spinal fluid (ACSF) composed of (in mM): 126 NaCl, 2.5 KCl, 1.25 NaH$_2$PO$_4$, 1 MgCl$_2$, 2 CaCl$_2$, 26 NaHCO$_3$, and 10 glucose (310 ± 5 mOs mL$^{-1}$, pH 7.4; bubbled with 95%O$_2$/5%CO$_2$) containing 0.2 mM ascorbic acid and 4 mM lactic acid at room temperature. The bath solution (ACSF) was continuously perfused in the recording chamber, where the temperature was adjusted to 30 ºC. Recordings were obtained in whole-cell mode, using pipettes filled with a solution containing (in mM): 130 potassium methyl-sulfate, 0.5 EGTA, 2 MgCl$_2$, 2 Na$_2$ATP, 0.2 GTP, 20 HEPES, 0.1 leupeptin and 0.75% biocytin (pH 7.2, 290 ± 5 mOs mL$^{-1}$). L5 PCs and interneuron cell-types were identified with their firing properties in response to current pulse injections[4,22]. To examine spiking activity during light stimulation, cell-attached recordings were obtained with the same pipettes as for whole-cell recordings. In the case of interneuron recordings, whole-cell recordings were obtained after cell-attached recordings to identify inhibitory cell subtypes. To activate ChR2, blue light illumination (482 nm for the maximum wavelength) was applied through an ×40 water immersion objective lens with an LED light system (Brainvision). Intensity and duration of light stimulation were controlled with a computer. To activate network activity, ramp-shaped light illumination, which had a slope of 0.1–0.2 mW s$^{-1}$ and started at zero intensity, was applied. In whole-cell recordings, membrane potentials were hyperpolarized by DC current injection during photostimulation to prevent spike discharge. In analysis of cross-correlation between membrane potentials of dual recordings, its significance of correlation was tested as follows. One of the traces was time-shifted from −30 to 30 ms at 2-ms step. We obtained correlation changes between the time-shifted trace and the other cell trace and calculated confident intervals (95%).

To identify PC subtypes in slice preparations, retrograde fluorescent tracers were injected in vivo into the contralateral cortex for IT cell labeling and into the ipsilateral pontine nuclei for PT cell labeling in P21–28 old rats anesthetized with ketamine (40 mg kg$^{-1}$, i.m.) and xylazine (4 mg kg$^{-1}$, i.m.), as described previously[6]. The injection coordinates of the contralateral frontal cortex and ipsilateral pontine nuclei were 4, 1.5–2.5, 0.5–0.8, and −5.6, 0.5–1, 9, respectively (anterior to bregma, lateral to bregma, depth, all in mm). Recordings were performed 2–3 days after tracer injections. Alexa Fluor 555-conjugated cholera toxin subunit B (Invitrogen) and rhodamine-labeled latex microspheres (RetroBeads$^{TM}$ Red, Lumafluor) were used to distinguish the two subtypes in the same preparation.

All data were obtained and analyzed using Axograph (AxoGraph). Data were represented as mean ± SD.

**Motor learning and optogenetic manipulation**. Motor learning was examined using a forced wheel running system as follows. Animals were made to run on foot bars (6 mm in diameter and 10 cm in length) attached to a custom-made wheel (35 cm in diameter), rotating at a constant speed (~10 cm s$^{-1}$) driven by a motor, in order to drink water from the supply port (Fig. 6a). The wheel had 110-foot bars that were aligned along the rim and removable from the wheel to make an arbitrary pattern of bar intervals. For optogenetic manipulation of PC activity, rats expressing ChR2-Venus or eArch-YFP with mCherry in L2/3, L5 IT, or L5 PT cells in bilateral motor areas were used.

The day before the task, the water supply was restricted. The weight of each animal was measured every day. If the weight was decreased, water was additionally given after trial sessions of the day before returning to their home cages where rats were allowed free access to dry pellet food. The running trial was continued for 3 min and performed five times a day (trial interval: at least 1 h). First, 4-week-old rats were habituated to running on the wheel consisting of foot bars aligned at a constant interval (~1 cm) for 4 days (regular pattern task). After that, fiber optic cannulas (core diameter, 0.4 mm; 0.39 NA) were implanted onto M1 forelimb areas, identified in our previous study[31], of both hemispheres. Three days after implantation, we let the rats start a pattern learning task in while they ran on the foot bars lined up repeatedly at the following intervals: 2, 6, 6, and 2 cm (Fig. 6b). One turn of the wheel included a section where foot bars aligned at regular intervals (length, 10 cm). To evaluate the progress of pattern learning, we counted the number of times the forelimbs touched the floor during a trial and normalized it to that in the first trial. LED light illumination (Thorlabs) was applied from the surface of the cortex through optical fibers connected to cannulas. To activate ChR2, ramp-shaped blue light (wavelength, 470 nm) was applied at 0.5 Hz (duration, 1 s; slope, 0.4 mW s$^{-1}$). To activate eArch, 1 mW green light (wavelength, 530 nm) was continuously illuminated during the task. Animals implanted optical cannulas without light stimulation were used as control.

**In vivo recordings**. Chronic in vivo recordings were obtained from the M1 forelimb area of Wistar rats aged 6–7 weeks, using silicon probe electrodes[67]. The electrode (NeuroNexus, A1x16-3mm-50-177) had 16 recording sites aligned line-arly and was mounted on a micromanipulator. Rats were anesthetized by isoflurane

(1–1.5%) during surgery. Insertion coordinates of the electrode were 1–1.5 mm anterior to bregma, 2–2.5 mm lateral to bregma, and 0.3 or 0.35 mm in depth. Two stainless steel screws inserted above the cerebellum were used as ground and indifferent electrodes, respectively. A Faraday cage was built surrounding the electrode to reduce noise during recordings. After surgery, the animals were recovered for at least one week before starting the task, including the regular task. The inserted electrode was moved ~50 μm a day in the vertical direction during recovery and the task period. We analyzed data obtained in L5, determined from the depth. Recording sites for L5 were positioned within 0.6–0.9 mm depth from the pia. We also analyzed data obtained from the recording sites positioned within 0.25–0.35 mm depth from the pia for activity in L2/3. All data were sampled at 20 kHz with Axoscope and analyzed with AxoGraph. LFPs were isolated by a band-pass filter. Power spectra of LFPs were calculated every 300 ms segments of LFPs and averaged. CSD of band-pass filtered LFPs was conducted to find the spatial location of the current source and sink[68].

**Computational simulation.** Both PC and FS cell models were constructed as three compartments (soma and two dendrites) with Hodgkin-Huxley-type conductances. The FS cell model used here has been previously described[27]. PC model was modified from Wang's model[69] and represented as a soma with proximal (d1) and distal (d2) apical dendritic compartments. The soma of the model cell had a sodium current ($I_{Na}$), potassium currents, calcium current ($I_{Ca}$), and a leak current ($I_l$). PCs show spike frequency adaptation in response to depolarizing current pulses, depending on the projection areas[4]. M type potassium current ($I_{MK}$) can be responsible for spike frequency adaptation in PCs[70,71]. Therefore, the soma compartment of the model contained two types of K$^+$ currents: a delayed rectifier K$^+$ current ($I_K$), which had a relatively higher activation threshold and faster activation time constant, and $I_{MK}$, which had a lower activation threshold and slower activation time constant. PT and IT type firing properties were reproduced by applying different $I_{MK}$ conductance values to each subtype. The proximal dendrite of the model had persistent sodium current ($I_{pNa}$), $I_{Ca}$, $I_K$, and $I_l$. The distal dendritic compartment had $I_{ca}$, a transient A type K$^+$ current ($I_A$), and $I_l$. The membrane potentials of the soma and dendrite of the PC model are described as follows:

$$C_m \frac{dV}{dt} = -I_{Na} - I_K - I_{MK} - I_{Ca} - I_l - I_{d1} \tag{1}$$

$$C_{md1} \frac{dV_{d1}}{dt} = -I_{pNa} - I_K - I_l - I_{soma} - I_{d2} \tag{2}$$

$$C_{md2} \frac{dV_{d2}}{dt} = -I_A - I_{Ca} - I_l - I_{d1} \tag{3}$$

where $C_m$, $C_{md1}$, and $C_{md2}$ are the membrane capacitances of the soma and of the proximal and distal dendrite, and were assigned values of 1, 1.5, and 0.5 μF/cm$^2$, respectively; $V$, $V_{d1}$, and $V_{d2}$ are, respectively, the membrane potentials of the soma and of the proximal and distal dendrite; and $I_{soma}$, $I_{d1}$, and $I_{d2}$ are the total currents of the soma and the two dendrites, respectively. Coupling conductance between d1 and d2 and between d1 and soma was set to 0.14 and 0.18 mS, respectively. Ionic currents in the model are given by the following Hodgkin-Huxley type equations:

$$I_{Na} = g_{Na} m^3 h (v - v_{Na}) \tag{4}$$

$$I_{pNa} = g_{pNa} a^3 (v - v_{Na}) \tag{5}$$

$$I_K = g_K n^4 (v - v_K) \tag{6}$$

$$I_{MK} = g_{MK} b (v - v_K) \tag{7}$$

$$I_A = g_A c^4 d (v - v_K) \tag{8}$$

$$I_{Ca} = g_{Ca} f^2 q (v - v_{Ca}) \tag{9}$$

$$I_l = g_l (v - v_l) \tag{10}$$

where $a$, $b$, $c$, $d$, $f$, $h$, $m$, $n$, and $q$ are activation and inactivation variables; $V_{Na}$, $V_K$, $V_{Ca}$, and $V_l$ are the reversal potentials of the sodium, potassium, calcium, and leak currents, respectively (in mV); and $g_{Na}$, $g_{pNa}$, $g_K$, $g_{MK}$, $g_A$, $g_{Ca}$, and $g_l$ are the maximal conductances (in mSc$^{-1}$m$^{-2}$). The gating kinetics of the ionic conductances are governed by equations of the following form:

$$\frac{dw}{dt} = \frac{w_\infty(v) - w}{\tau_w} \tag{11}$$

where $w$ stands for one of $a$, $b$, $d$, $f$, $h$, $m$, $n$, and $q$, and the steady-state activation and inactivation functions are given by

$$w_\infty = \frac{1}{1 + \exp[(v - \theta_w)/k_w]} \tag{12}$$

where $\theta_w$ and $k_w$ are the half-activation/half-inactivation voltage and slope, respectively. Parameter values used in simulations are shown in supplementary Table 1. Time constant ($\tau_w$ in ms) for $I_{Na}$ activation is given by

$\tau_m = 0.5 + 0.14 * \exp(-(v + 25.5)/10)$. When V is less than −25.5 mV, $\tau_m$ is replaced by $1.8 + 0.14 * \exp((v + 25.5)/10)$. The inactivation time constant is given by $\tau_h = 0.15 + 11.15/(1 + \exp(v + 33.5)/15)$. $I_{pNa}$ activation time constant $\tau_a$ is given by $2.5 + 0.145 * \exp(-(v + 40)/10)$. When $V$ is less than −40 mV, $\tau_a$ is replaced by $2.5 + 0.145 * \exp((v + 40)/10)$. $I_K$ activation time constant is given by $\tau_n = 2.5 + 4.35 * \exp(-(v + 10)/10)$. When $V$ is less than −10 mV, $\tau_n$ is replaced by $10 + 4.35 * \exp((v + 10)/10)$. Activation time constant for $I_{MK}$ is given by $\tau_b = 3300/(\exp((v + 35)/20) + \exp((-(v + 35)/20))$. $I_A$ activation time constant is given by $\tau_c = 0.2 + 25/(\exp((v + 60)/5) + \exp((-(v - 3)/55))$. When V is less than −65 mV, $\tau_c$ is assigned constant values of 1.0 ms. $I_A$ inactivation, and $I_{Ca}$ activation and inactivation time constants were assumed as constant value of 25, 4, and 60 ms, respectively.

L5 model circuits were constructed from 400 PCs and 50 FS model cells. PCs were divided into an equal number of PT and IT type cells. A random distribution of cells within a square, measuring 500 μm on each side, was assumed. The coordinates of the cells were determined by random numbers from 1 to 500. Model cells were assumed to have a probability of forming synaptic connections with cells located within 150 μm. Postsynaptic cells were randomly selected from candidates with a connection probability observed in slice experiments. Connection patterns including connection probabilities from IT to IT[7,8], PT to PT[8], IT to PT[7], FS to FS[27], and between PC and FS cells[22,27] are shown in Fig. 5b. In the circuit model, electrical synapses were formed between the dendritic compartments of FS cells. In our slice experiments, reciprocal chemical synaptic connections were frequently observed only in electrically connected FS cell pairs[27]. Therefore, no reciprocal chemical connections were formed between electrically unconnected FS cells.

For chemical synaptic inputs, we assumed that transmitters were released in the cleft for 1.4 ms when the membrane potential of the presynaptic cell overshot the threshold potential (0 mV). During this period, the postsynaptic conductance change is given by the following equation:

$$\frac{dr}{dt} = \alpha(1 - r) - \beta r \tag{13}$$

where $\alpha$ and $\beta$ are the binding and unbinding rate of transmitters. Transmitters are then removed from the cleft. The postsynaptic conductance change for this period is given by the following equation:

$$\frac{dr}{dt} = -\beta_2 r \tag{14}$$

where $\beta_2$ is the removal rate of transmitters. For excitatory connections between PCs, $\alpha$, $\beta$, and $\beta_2$ were set to 0.7, 0.3, and 0.18, respectively. Those of excitatory connections to FS cells were set to 0.8, 0.2, and 0.5, respectively. Those inhibitory connections to PCs were set to 0.35, 0.1, and 0.08, respectively. Those synaptic connections between FS cells were set to 0.5, 0.2, and 0.15, respectively. Synaptic inputs between PT cells show short-term facilitation in response to repetitive spiking activities[8]. By contrast, synaptic connections from IT to IT, IT to FS, and inhibitory connections from FS cells show short-term depression[8,27,43]. In the model, the scale factor was included as short-term plasticity and multiplied by the synaptic conductance[72]. During the transmitter release period, the scale factor change is given by the following equation:

$$\frac{ds}{dt} = (1 - \alpha_s s)/\beta_s \tag{15}$$

where $\alpha_s$ and $\beta_s$ are the rates of the scale factor. The scale factor, then, returns to 1. Scale factor change for this period is given by the following equation:

$$\frac{ds}{dt} = (1 - s)/\beta_{s2} \tag{16}$$

where $\beta_{s2}$ is the recovery rate of the scale factor. In the scale factor for facilitation, we assumed $\alpha_s$, $\beta_s$, and $\beta_{s2}$ were 32, −145, and 150, respectively. For the depression type synapse, we assumed $\alpha_s$, $\beta_s$, and $\beta_{s2}$ were 32, 145, and 150, respectively. The maximal and minimal scale factors were set to 1.2 and 0.3, respectively. The reversal potential of excitatory synapses in PCs and FS cells was set to 0 mV. The reversal potential of inhibitory synapses in PCs was set to −80 mV. We set the reversal potential of inhibitory synapses in FS cells as −58 mV, which was experimentally determined in cortical FS cells using gramicidin-perforated patch-clamp recordings[73]. Synaptic delay was set to 1 ms. Chemical excitatory and inhibitory synapses between L5 model cells were assumed to form on the soma compartment. For simplicity, we used constant values for the maximum excitatory and inhibitory synapse conductance in PCs and FS cells, which were set at 0.14 and 0.33 mS in PCs, and 0.2 and 0.3 mS in FS cells, respectively. To activate network activity in the model circuits, excitatory synaptic conductance, as external inputs, was added to the proximal dendrite of PCs and the soma of FS cells. External synaptic inputs in individual cells were randomly activated. The average input frequency of external inputs to PT, IT, and FS cells was set to 5, 2.5, and 5 Hz, respectively, reflecting a connection probability from L2/3 to L5 cells observed in our experiments[4,22]. Because L5 PCs receive common synaptic inputs from L2/3, depending on the pyramidal subtypes, common inputs to PT or IT cells were included in external inputs of the model. As common inputs, we assumed the five sources for PT and IT cells, respectively. Each PT cell received inputs on the proximal dendritic compartment from one of the five sources. Similarly, each IT cell received inputs from one of another five sources. Common inputs were also

randomly activated at an average input frequency of 5 and 2.5 Hz, reflecting a common input probability observed in slice experiments, in PT and IT cells, respectively. The conductance of external synaptic inputs was set to 0.22 mS.

All simulations reported here were performed using Visual Studio (Microsoft). Programs were written in C language. Differential equations were solved by a fourth-order Runge-Kutta algorithm (time step, 0.01 ms).

**Statistics and reproducibility**. Data were statistically analyzed by KaleidaGraph (version 4.1J, Synergy software) and Excel (Microsoft Office 2019). Error bars represent SD. Center line and box limits of box plot indicate median and upper/lower quartiles, respectively. Whiskers of box plot represent 1.5× interquartile range. Nonparametric and parametric data comparisons were performed using one-way ANOVA with Bonferroni post hoc tests, Wilcoxon signed-rank test, Welch's t-test, and paired t-test. Significance was accepted at $P < 0.05$. All electrophysiological, behavioral, and immunohistological experiments were obtained from several animals. In slice experiments, the same protocol was repeatedly applied 5–10 times to the recording cell. Simulation results were confirmed with multiple trials.

**Reporting summary**. Further information on research design is available in the Nature Research Reporting Summary linked to this article.

## Data availability
Essential source data were uploaded with the manuscript as Supplementary Data 1–10. Additional source data are available from the corresponding author upon reasonable request.

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

## Acknowledgements

We thank Kenji Mizuseki and Nobuyoshi Matsumoto for advice about chronic in vivo recordings and Yumiko Hatanaka for advice about electroporation procedure. This work was supported by JSPS KAKENHI: 25115730, 16K07013, 17H06311, and 20H03359.

## Author contributions

T.O. designed and performed all experiments and simulations, analyzed data, and wrote the manuscript. Y.K. discussed and wrote the manuscript.

## Competing interests

The authors declare no competing interests.
