## [Peer Review File · Communications Biology]

Reviewers' Comments:

Reviewer #1:

Remarks to the Author:

This study, by Otsuka and Kawaguchi, uses multiple approaches to address the circuit mechanisms through which motor cortical circuits produce beta/gamma frequency oscillations and examine how such activities are related to motor learning and execution. This is undoubtedly an important issue, both in the fields of cortical physiology and pathophysiology. While interactions between pyramidal and fast spiking/PV neurons have been identified as being crucial in generating these activities in other brain areas (most notably hippocampus), to my knowledge this is not the case for motor cortex. A particular strength of the study is that the investigators are able to show matching changes to oscillatory activity both in membrane potentials from slice recordings and from LFPs recorded in vivo to the same perturbation. This validates the importance of the later, by suggesting that the association between LFP activities and behaviour reflect specific cellular activities. While the importance of PV neurons in generating these activities is not surprising based on other studies of cortex and hippocampus, of particular importance is that the authors are able to delineate the role of Layer V IT and PT pyramidal neurons. Their approach of labelling these neurons via injection of retrograde tracers/in utero electroporation may be more cumbersome than using transgenic mice, but should result in unambiguous labelling/transfection of these populations and provides validation in a species on which much cortical anatomy and physiological has been elucidated. While the study has many strengths, clarification and elaboration on certain aspects of the study could strengthen and validate the authors claims.

1. The authors show an association between changes in the frequency of cortical LFP content and learning. However, there are obvious confounds in that the manipulations they use to change the oscillatory content will also change the overall output of different populations of cortical neurons (i.e. driving cortical layer 2/3 neurons will alter the firing rate of many cortical populations irrespective of what is happening to the LFP). I have sympathy that it is difficult to completely delineate the contributions of oscillation and firing rate with any intervention. Having said that, the authors could explore their data more thoroughly to test the strength of the association. Why is the analysis of LFPs in relation to behaviour over trials restricted to Day1 and Day3? Do the beta/gamma LFP timecourses across trials correlated with the behavioural full behavioural profile (Fig 6C) or just with the initial trials? And does the 60Hz gamma activity remain flat over the learning period? Answering these questions would give a much richer view of how oscillations in motor cortex are associated with motor learning and strengthen the current claims of the authors.
2. The method of transfecting specific populations of neurons using electroporation is elegant in allowing the investigators to use rats, rather than transgenic mouse lines. The cell counts using C-TIP2 immunohistochemistry to show that their method can divide PT and IT neurons are convincing, but little data is shown to demonstrate that the transfection is layer specific (in any of the experiments). The authors should show this, particularly in relation to the optogenetic in vivo experiments where non-specific opsin expression would be a significant confound.
3. In their current form, I find it difficult to fully evaluate the robustness of the changes in oscillatory power for the in vivo experiments (Fig. 6). One or more of the following could address this a) error bars on the power spectra in 6e. b) Comparison of Layer V LFPs to those recorded from other Layers (Figure 6 legend should state that LFPs are Layer V only). c) It would also be useful to know if there are consistent changes in absolute power within animals (i.e. using a paired test), rather than always looking at ratios. If the comparison is matched within animal, gross changes in baseline power (e.g. caused by electrode impedance) should be relatively small
4. Movement artifacts can corrupt analyses of LFPs in behaving animals. The similar nature of behaviour across conditions should partly control for this, can the authors provide further evidence that such artifacts were not a major problem in their recordings?

5. The n-numbers for LFP analysis are higher than for behaviour – did only some of the animals undergo behavioural evaluation or are multiple recordings from the same animals included in the LFP analysis?

6. The authors state that “These results suggest that PT cells, but not IT cells, are involved in the generation of oscillatory activity”. This is not correct in two ways. Firstly, the authors findings relate only to oscillations generated through the stimulation of Layer 2/3 neurons. IT cells could be engaged in oscillations through a different circuit mechanism. Secondly, these findings do not necessarily relate to oscillations per se. Indeed, there is significant power at lower and higher frequencies that presumably are generated through different circuit mechanisms. The authors should take greater care that statements regarding relation between the circuits they are manipulating and specific activities.

7. The data on the role of gap junctions (Supp. Fig 2) don’t appear to have any statistical analysis – this should be added or they should be removed.

8. The analysis describing membrane potential correlations in figure 2 are potentially very important. However, it would be beneficial to add analysis showing that the correlations are significant per se, before analysing the time lag (as without this the time lag could be meaningless). This could be achieved by comparing the “real” correlation to time shifted data, which should not be correlated due to drift in the phase and frequency of the ongoing oscillation.

Reviewer #2:

Remarks to the Author:

This is an impressively expansive approach to understanding cortical microcircuitry and their communication and their link to behavior. The authors use slice physiology from multiple animals with varying labelling of cell types, simulation and behavioral interventions. The results should be a nice addition to the literature and should be of broad interest. While in general, I am impressed by this manuscript and its scope, there are several methodological issues that require clarification. The main issue in my opinion is the link between the slice physiology and the link to behavior/LFP (and its interpretation).

1. For the laminar recordings, how was layer 5 identified? It might be really helpful to show CSD analysis? If I understand the approach, this is more principled than simply referencing relative to the cerebellum. It can also identify layer specific info as well as ensuring there is no volume conduction.

2. Little layer specific staining data is shown regarding the validation of the electroporation, dating of embryos and the cell types. Figures showing this more clearly would help.

3. The LFP analysis is less convincing. Raw traces are not shown. Filtering should be applied on top of raw.

4. How did they account for movement artifacts?

5. Fig 6e shows that the greatest power is in theta (also see below) . The change in ~30Hz could be a harmonic of a lower frequency. This can also be due to a change in the shape of the waveform. Is there any evidence of this?

6. The authors cite mainly non-human primate literature. There is a relatively sparse lit on rodents and beta/learning. In fact, there is more of a link to theta. There might be species specific differences. Given the link of gamma to theta, and the more prevalent data for theta with learning, what is the authors model of this? Example refs

a. <https://pubmed.ncbi.nlm.nih.gov/22388818/>

b. <https://pubmed.ncbi.nlm.nih.gov/31133689/>

7. Fig 7a/b - raw traces and the full spectrum should be shown.

8. Beta has been linked to an “antikinetic” state; there is little discussion of the gamma vs beta findings. This should be expanded.

Minor.

- Fig 1e: it is not clear what the traces show. Membrane potential? LFP?
- The faster learning is intriguing. It is worth discussing in terms of possible mechanisms.

Reviewer #3:

Remarks to the Author:

In this work by Dr Otsuka et al., the authors well demonstrated that the beta/gamma oscillation induced by optogenetic manipulation of cortical PT-FS network promotes motor learning. In specific, they firstly identified different pyramidal cell types in motor cortex by doing in utero electroporation at different embryonic days. By doing ramp-shaped light illumination in a cell type specific manner, they found oscillations ranging in the beta/gamma frequency band were induced in vitro, specific to L5 PT cells but not IT cells. And what's more important, oscillations in a similar frequency band were successfully observed in LFPs in vivo by optogenetic manipulating PT-FS cell subnetworks, which were proved to play crucial roles in the progression of motor learning process. Overall, this manuscript is well written and the experiments results are well described. I recommend that this paper to be accepted after minor revision, and I have the following comments and questions to help improving the quality of the manuscript.

1. Fig1d: In legend, please describe the meaning of each symbol (i.e. red triangle, grey triangle, grey circle, hollow triangle). To general readers, this is important to understand the local circuit between pyramidal neurons and interneurons.
2. Fig 1f: How did author identify PCs and FS cells in L5? As I understand, only L2/3 PCs were visualized with Chr2-Venus, so based on what property (cell morphology?), were they able to differentiate L5 PCs from FS cells?
3. Fig 1d-e: for the trace example being shown, was it recorded from L5 PC or FS cell? Please clarify this in legend.
4. In Results paragraph 1, it's described that "During ramp-shaped light illumination, oscillatory membrane activity was observed in L5 PCs, as well as in FS cells". Is this due to the shared input from L2/3 PCs reaching L5 PCs and FS cells, or the localized reciprocal connection between L5 PCs and FS cells? I'm curious whether L2/3 PCs also innervate L5 FS cells directly? This point also relates to the simulation network shown in Fig 5b and schematic drawing in Fig 8d.
5. Fig 2: how about the correlation level between PC/PC pair and PC-FS pair without optogenetics? That will be a good control to value the significance of light stimulation effect.
6. Fig 3a & Fig 4a: Is it possible to quantify the colocalization level of PT/IT cells labelled by retrograde fluorescent tracer and in utero electroporation? General readers may get confused why the author chose two different methods to identify L5 PC subtypes.
7. Fig 5b: Please explain or show reference in supporting the connection property between IT and PT cells (why $p = 0.1$?).
8. Fig 6: The author stated that the beta/gamma band oscillation was related to the learning process but not locomotive behaviour itself. I'm curious about the running speed in early and late learning phase (day1 vs day3). This evidence can further argue whether the power of beta/gamma partially correlates to movement variable, like speed.
9. Fig 6h & Fig 7d: As the beta/gamma band activity (25-45Hz) was necessary for motor learning, I don't understand why the power of beta/gamma was decreased as the learning progressed on day3 (Fig 6h). According to Fig 7d, the Chr2 photo stimulation enhanced beta/gamma oscillation and therefore facilitated learning progress. I expect the power of beta/gamma in late learning phase should be higher. Do I misunderstand the data?
10. In the method, paragraph 2 of Slice experiments part, please verify for the retrograde fluorescent tracing experiment, how old were the animals?

Response to reviewer

Reviewer #1 (Remarks to the Author):

This study, by Otsuka and Kawaguchi, uses multiple approaches to address the circuit mechanisms through which motor cortical circuits produce beta/gamma frequency oscillations and examine how such activities are related to motor learning and execution. This is undoubtedly an important issue, both in the fields of cortical physiology and pathophysiology. While interactions between pyramidal and fast spiking/PV neurons have been identified as being crucial in generating these activities in other brain areas (most notably hippocampus), to my knowledge this is not the case for motor cortex. A particular strength of the study is that the investigators are able to show matching changes to oscillatory activity both in membrane potentials from slice recordings and from LFPs recorded in vivo to the same perturbation. This validates the importance of the later, by suggesting that the association between LFP activities and behaviour reflect specific cellular activities. While the importance of PV neurons in generating these activities is not surprising based on other studies of cortex and hippocampus, of particular importance is that the authors are able to delineate the role of Layer V IT and PT pyramidal neurons. Their approach of labelling these neurons via injection of retrograde tracers/in utero electroporation may be more cumbersome than using transgenic mice, but should result in unambiguous labelling/transfection of these populations and provides validation in a species on which much cortical anatomy and physiological has been elucidated. While the study has many strengths, clarification and elaboration on certain aspects of the study could strengthen and validate the authors claims.

1. The authors show an association between changes in the frequency of cortical LFP content and learning. However, there are obvious confounds in that the manipulations they use to change the oscillatory content will also change the overall output of different populations of cortical neurons (i.e. driving cortical layer 2/3 neurons will alter the firing rate of many cortical populations irrespective of what is happening to the LFP). I have sympathy that it is difficult to completely delineate the contributions of oscillation and firing rate with any intervention. Having said that, the authors could explore their data more thoroughly to test the strength of the association. Why is the analysis of LFPs in relation to behaviour over trials restricted to Day1 and Day3? Do the beta/gamma LFP timecourses across trials correlated with the behavioural full behavioural profile (Fig 6C) or just with the initial trials? And does the 60Hz gamma activity remain flat over the learning period? Answering these questions would give a much richer view of how oscillations in motor cortex are associated with motor learning and strengthen the current claims of the authors.

Following the suggestion, we have analyzed LFP data obtained at day2 during learning period. The beta/gamma oscillation power progressively became weaker through the trial days. By contrast, theta and 60-70 Hz frequency band activity were preserved. We have included these results to figure 6 (Fig. 6h).

2. The method of transfecting specific populations of neurons using electroporation is elegant in allowing the investigators to use rats, rather than transgenic mouse lines. The cell counts using C-TIP2 immunohistochemistry to show that their method can divide PT and IT neurons are convincing, but little data is shown to demonstrate that the transfection is layer specific (in any of the experiments). The authors should show this, particularly in relation to the optogenetic in vivo experiments where non-specific opsin expression would be a significant confound.

Following the suggestion, we have analyzed distribution of cells labeled by in utero electroporation at E14.5 and E15.5. In both cases, more than 80 % of labeled cells were located within L5. Labeled cells at E15.5 were located in the upper L5 more than those in lower L5. By contrast, proportion of labeled cells at E14.5 located in lower L5 were increased. We have included these results as a supplementary figure (supplementary Fig. 3; text, page7, line 31 to page 8, line 5).

3. In their current form, I find it difficult to fully evaluate the robustness of the changes in oscillatory power for the in vivo experiments (Fig. 6). One or more of the following could address this a) error bars on the power spectra in 6e. b) Comparison of Layer V LFPs to those recorded from other Layers (Figure 6 legend should state that LFPs are Layer V only). c) It would also be useful to know if there are consistent changes in absolute power within animals (i.e. using a paired test), rather than always looking at ratios. If the comparison is matched within animal, gross changes in baseline power (e.g. caused by electrode impedance) should be relatively small

- a) Following the suggestion, we have changed Fig. 6e to the power spectra of LFPs represented as mean \pm SEM. Accordingly, we have deleted Fig. 6g (in previous figure) that showed the power ratio of beta/gamma band during regular and patterned task and the rest state.
- b) We analyzed LFPs data obtained in L2/3 during learning trial (day1). We detected theta and gamma (~60 Hz) frequency band activity during trial, similar to those in L5. However,

no obvious increase of beta/gamma frequency band was observed. We have included the power spectra of LFPs obtained in L2/3 during pattern learning into figure 6 (Fig. 6f; text, page10, line23-28).

- c) Following the suggestion, we compared absolute peak power of theta, beta/gamma, and 60-70 Hz frequency band through learning days (Fig. 6h). Only beta/gamma band was decreased through learning trial days.

4. Movement artifacts can corrupt analyses of LFPs in behaving animals. The similar nature of behaviour across conditions should partly control for this, can the authors provide further evidence that such artifacts were not a major problem in their recordings?

We examined whether wheel rotating speed affects LFPs. During the trial, wheel rotating speed was changed from 10 to 7 and then to 14 cm/sec. Consistent with previous study (von Nicolai et al., J Neurosci., 2014; ref 37), the peak and frequency of theta and gamma (~60 Hz) band activity was changed, depending on the rotating speed. However, beta/gamma band activity was preserved. We have included these results as a supplementary figure (Supplementary Fig. 6; text, page10, line 29 – page 11, line 1).

5. The n-numbers for LFP analysis are higher than for behaviour – did only some of the animals undergo behavioural evaluation or are multiple recordings from the same animals included in the LFP analysis?

For learning evaluation, we only used animals implanted optical cannulas. During learning task, light illumination was continuously applied through optical fibers to animals expressing ChR2 or eArch. For experiments of recordings with light stimulation, we had intervals without light stimulation to examine the effect of light stimulation on LFPs. Therefore, we did not include data obtained in animals with LFPs recordings for learning evaluation. In addition, we confirmed learning time courses in animals with electrode inserted and in animals implanted optical cannulas (Fig. 6c). No significant difference was found between these animals. We have included these results as a supplementary figure (Supplementary Fig. 4). We also added the description about animals that used for Fig. 6c (figure legend).

6. The authors state that “These results suggest that PT cells, but not IT cells, are involved in the generation of oscillatory activity”. This is not correct in two ways. Firstly, the authors findings relate only to oscillations generated through the stimulation of Layer 2/3 neurons. IT cells could be engaged in oscillations through a different circuit mechanism. Secondly, these

findings do not necessarily relate to oscillations per se. Indeed, there is significant power at lower and higher frequencies that presumably are generated through different circuit mechanisms. The authors should take greater care that statements regarding relation between the circuits they are manipulating and specific activities.

Following the suggestion, we have deleted “but not IT cells” from the text (page 6, line 25).

7. The data on the role of gap junctions (Supp. Fig 2) don't appear to have any statistical analysis – this should be added or they should be removed.

Following the suggestion, we have examined statistical analysis between samples. Statistical significance was included into the figure (Supplementary Fig. 2).

8. The analysis describing membrane potential correlations in figure 2 are potentially very important. However, it would be beneficial to add analysis showing that the correlations are significant per se, before analysing the time lag (as without this the time lag could be meaningless). This could be achieved by comparing the “real” correlation to time shifted data, which should not be correlated due to drift in the phase and frequency of the ongoing oscillation.

Following the suggestion, we examined whether the cross-correlation was significant by correlation analysis applied to time shifted data. One of traces was shifted from -30 to 30 msec at 2-msec step. We then obtained cross-correlation between time shifted trace and the other cell trace and calculated confident intervals (95%) for correlation. We confirmed the significance of cross correlation of cell pairs. We have included these descriptions to the method and added 95% confident intervals to the figure (Fig. 2; text, page 18, line 2-6).

Reviewer #2 (Remarks to the Author):

This is an impressively expansive approach to understanding cortical microcircuitry and their communication and their link to behavior. The authors use slice physiology from multiple animals with varying labelling of cell types, simulation and behavioral interventions. The results should be a nice addition to the literature and should be of broad interest. While in general, I am impressed by this manuscript and its scope, there are several methodological issues that require clarification. The main issue in my opinion is the link between the slice physiology and the link to behavior/LFP (and its interpretation).

1. For the laminar recordings, how was layer 5 identified? It might be really helpful to show CSD analysis? If I understand the approach, this is more principled than simply referencing relative to the cerebellum. It can also identify layer specific info as well as ensuring there is no volume conduction.

We determined L5 recordings by the position of the electrode from the pia. In our analysis, recording sites for L5 were positioned between 0.6 and 0.9mm depth from the pia. Moreover, we analyzed data obtained in L2/3 to examine layer specificity. Recording sites were positioned between 0.25 and 0.35mm depth from the pia. In contrast to L5, the beta/gamma frequency band activity was weak in L2/3 recordings (Fig. 6f). We also examined CSD analysis for LFPs obtained at L5 and found sink and source components (supplementary Fig. 5). These results suggest that beta/gamma band activity is generated within L5 local circuits. We have included these results (supplementary Fig. 5) and the description about the position of recording site for analysis (page 10, line 25-28 and page 19, line 31 - page 20, line 1 and page 20, line 4-6).

2. Little layer specific staining data is shown regarding the validation of the electroporation, dating of embryos and the cell types. Figures showing this more clearly would help.

Following the suggestion, we have analyzed distribution of labeled cells by in utero electroporation at E14.5 and E15.5. In both cases, more than 80 % of labeled cells were located within L5. Labeled cells at E15.5 were located in the upper L5 more than those in lower L5. By contrast, proportion of labeled cells at E14.5 located in lower L5 were increased. We included these results as a supplementary figure (supplementary Fig. 3; text, page 7, line 31 – page 8, line 5). We also added the boarder of layer to the figure (Fig. 1c).

3. The LFP analysis is less convincing. Raw traces are not shown. Filtering should be applied on top of raw.

Following the suggestion, we have included raw traces into fig. 6, 7, and 8.

4. How did they account for movement artifacts?

We examined whether wheel rotating speed affects to LFPs. During the trial, wheel rotating speed was changed from 10 to 7 and then to 14 cm/sec. Consistent with previous study (von

Nicolai et al., J Neurosci., 2014; ref 38), the peak and frequency of theta and gamma (~60 Hz) band activity depended on the rotating speed. On the other hand, that of beta/gamma band activity was not changed. We have included these results as a supplementary figure (Supplementary Fig. 5, text, page 10, line 29 – page 11, line 1).

5. Fig 6e shows that the greatest power is in theta (also see below). The change in ~30Hz could be a harmonic of a lower frequency. This can also be due to a change in the shape of the waveform. Is there any evidence of this?

We have changed fig. 6e to the power spectra represented as mean \pm SEM. The beta/gamma band activity was selectively observed during learning task, although similar power of theta band activity was observed during both regular and pattern learning tasks.

6. The authors cite mainly non-human primate literature. There is a relatively sparse lit on rodents and beta/learning. In fact, there is more of a link to theta. There might be species specific differences. Given the link of gamma to theta, and the more prevelant data for theta with learning, what is the authors model of this? Example refs

a. <https://pubmed.ncbi.nlm.nih.gov/22388818/>

b. <https://pubmed.ncbi.nlm.nih.gov/31133689/>

Following the suggestion, we have cited the papers described above and discussed about theta and beta/gamma wave in learning (p14, line12-17).

7. Fig 7a/b - raw traces and the full spectrum should be shown.

Following the suggestion, we have included raw trace to the figure 7a. In addition, we have changed to full power spectra of LFPs.

8. Beta has been linked to an “antikinetic” state; there is little discussion of the gamma vs beta findings. This should be expanded.

Following the suggestion, we have included the description about gamma band to the text (page 14, line17-21).

Minor.

• Fig 1e: it is not clear what the traces show. Membrane potential? LFP?

Fig. 1e is magnified traces of membrane potentials of the cell shown in Fig. 1d. We have modified the legend of figure 1 to understand clearly.

- *The faster learning is intriguing. It is worth discussing in terms of possible mechanisms.*

Following the suggestion, we have added the description in the text (page 14, line 3-11).

Reviewer #3 (Remarks to the Author):

In this work by Dr Otsuka et al., the authors well demonstrated that the beta/gamma oscillation induced by optogenetic manipulation of cortical PT-FS network promotes motor learning. In specific, they firstly identified different pyramidal cell types in motor cortex by doing in utero electroporation at different embryotic days. By doing ramp-shaped light illumination in a cell type specific manner, they found oscillations ranging in the beta/gamma frequency band were induced in vitro, specific to L5 PT cells but not IT cells. And what's more important, oscillations in a similar frequency band were successfully observed in LFPs in vivo by optogenetic manipulating PT-FS cell subnetworks, which were proved to play crucial roles in the progression of motor learning process.

Overall, this manuscript is well written and the experiments results are well described. I recommend that this paper to be accepted after minor revision, and I have the following comments and questions to help improving the quality of the manuscript.

1. Fig1d: In legend, please describe the meaning of each symbol (i.e. red triangle, grey triangle, grey circle, hollow triangle). To general readers, this is important to understand the local circuit between pyramidal neurons and interneurons.

Following the suggestion, we have added the description of inset to the figure legend (Fig. 1d).

2. Fig 1f: How did author identify PCs and FS cells in L5? As I understand, only L2/3 PCs were visualized with ChR2-Venus, so based on what property (cell morphology?), were they able to differentiate L5 PCs from FS cells?

We identified L5 PCs and FS cells with their firing properties. We modified the method section and sited our previous studies (page, 17, line, 22-23).

3. *Fig 1d-e: for the trace example being shown, was it recorded from L5 PC or FS cell? Please clarify this in legend.*

In figure 1d-e, we showed an example of recording from L5 PC. Following the suggestion, we have modified the figure legend (Fig. 1) to understand clearly.

4. *In Results paragraph 1, it's described that "During ramp-shaped light illumination, oscillatory membrane activity was observed in L5 PCs, as well as in FS cells". Is this due to the shared input from L2/3 PCs reaching L5 PCs and FS cells, or the localized reciprocal connection between L5 PCs and FS cells? I'm curious whether L2/3 PCs also innervate L5 FS cells directly? This point also relates to the simulation network shown in Fig 5b and schematic drawing in Fig 8d.*

In our previous study (ref. 22), we showed that connection probability from L2/3 PCs to L5 FS cell was approximately 10%. Common input probability from L2/3 PCs to L5 PC and FS cell pair was approximately 2%, that is a chance level for the connection probability from L2/3 to L5 PC/FS cell pair, and was independent of connection pattern between L5 PC and FS cell. By contrast, L5 FS cells frequently formed reciprocal connections with L5 PCs. Taken together with the observation in Fig. 2, these connection patterns suggest that oscillatory activity is generated at local circuits between L5 PCs and FS cells rather than common inputs from L2/3 to L5 PC/FS cell. We have added the description about connection pattern of PC/FS cells into the text (page 6, line 6-11).

5. *Fig 2: how about the correlation level between PC/PC pair and PC-FS pair without optogenetics? That will be a good control to value the significance of light stimulation effect.*

Following suggestion, we analyzed spontaneous excitatory synaptic currents (sEPSCs) onto L5 PC and FS cells in slice preparations. PC and FS cells received sEPSCs at 3.02 ± 1.18 and 12.83 ± 2.2 Hz, respectively. In PC/PC and PC/FS cell pairs, synchronized sEPSCs, simultaneously occurred within 5 msec, were rarely found at 0.05 ± 0.02 and 0.14 ± 0.05 Hz. We have included these statements into the text (page, 5, line, 23-27).

6. *Fig 3a & Fig 4a: Is it possible to quantify the colocalization level of PT/IT cells labelled by retrograde fluorescent tracer and in utero electroporation? General readers may get confused why the author chose two different methods to identify L5 PC subtypes.*

It is difficult to identify L5 PC subtypes with retrograde tracers in animals labeled L5 PC by *in utero* electroporation, because mCherry was simultaneously expressed to identify electroporated area. Therefore, we used identification of L5 PC subtypes by their firing properties that correlate with their projection areas (ref. 4, 5, 6). Following suggestion, we have added the statement for the reason why we used different identification methods (page, 8, line, 16-17).

7. Fig 5b: Please explain or show reference in supporting the connection property between IT and PT cells (why $p = 0.1$?).

In our previous study, connection probability from crossed-cortico-striatal (IT) to cortico-pontine (PT) cell was approximately 10 % (11/98 cell pairs) (ref. 7). We modified the text to easily find the references used in our model (page 22, line 15-16).

8. Fig 6: The author stated that the beta/gamma band oscillation was related to the learning process but not locomotive behaviour itself. I'm curious about the running speed in early and late learning phase (day1 vs day3). This evidence can further argue whether the power of beta/gamma partially correlates to movement variable, like speed.

Following the suggestion, we examined whether wheel rotating speed affects to LFPs. During the trial at day1, wheel rotating speed was changed from 10 to 7 and then to 14 cm/sec (duration, 1 min for each). Consistent with a previous study (von Nicolai et al., J Neurosci., 2014; ref 37), the peak and frequency of theta and gamma (~60 Hz) band activity was changed, depending on the rotating speed. However, the beta/gamma band activity was preserved. We have included these results as a supplementary figure (Supplementary Fig. 4, text, page 10, line 29 – page 11, line 1).

9. Fig 6h & Fig 7d: As the beta/gamma band activity (25-45Hz) was necessary for motor learning, I don't understand why the power of beta/gamma was decreased as the learning progressed on day3 (Fig 6h). According to Fig 7d, the ChR2 photo stimulation enhanced beta/gamma oscillation and therefore facilitated learning progress. I expect the power of beta/gamma in late learning phase should be higher. Do I misunderstand the data?

In our results, beta/gamma oscillation was progressively decrease through learning trial days, reflecting learning progress (Fig. 6c and h). Moreover, manipulation of beta/gamma

activity affected learning time courses (Fig. 7 and 8). These results suggest that beta/gamma oscillation play important roles in learning progress. However, after learning, beta/gamma band activity would be not necessary to execute learned behavior. We have modified the text (page 14, line 3-5).

10. In the method, paragraph 2 of Slice experiments part, please verify for the retrograde fluorescent tracing experiment, how old were the animals?

Following the suggestion, we have included age of the animals that we used for injection into the method (page 18, line 9).

Reviewers' Comments:

Reviewer #1:

Remarks to the Author:

The authors have satisfactorily addressed all my concerns.

Reviewer #2:

Remarks to the Author:

The authors responded to my comments appropriately. The authors should be congratulated on a very nice contribution.

Reviewer #3:

Remarks to the Author:

I went through all the materials that the authors provided, I am satisfied with their responses to all the comments that the reviewers raised. Therefore, I think the work is acceptable with its present form.